# Causal Matrix Completion under Multiple Treatments via Mixed Synthetic Nearest Neighbors

**Minrui Luo** [3]   **Zhiheng Zhang**[✉ 1 2]

## Abstract

Synthetic Nearest Neighbors (SNN) provides a principled solution to causal matrix completion under missing-not-at-random (MNAR) by exploiting local low-rank structure through fully observed anchor submatrices. However, its effectiveness critically relies on sufficient data availability within each treatment level, a condition that often fails in settings with multiple or complex treatments. In this work, we propose Mixed Synthetic Nearest Neighbors (MSNN), a new entry-wise causal identification estimator that integrates information across treatment levels. We show that MSNN retains the finite-sample error bounds and asymptotic normality guarantees of SNN, while enlarging the effective sample size available for estimation. Empirical results on synthetic and real-world datasets illustrate the efficacy of the proposed approach, especially under data-scarce treatment levels. [1]

## 1. Introduction

Causal inference from observational data has become a cornerstone of modern data science, enabling the rigorous evaluation of interventions across diverse domains such as economics, public policy, and digital platforms. A critical challenge in this field is the estimation of counterfactual outcomes—i.e., what would have transpired had a different treatment been applied—when faced with incomplete data and complex treatment structures. Recent advancements in matrix completion, a well-established tool from machine learning, have been adapted to this causal inference framework (Ma & Chen, 2019; Agarwal et al., 2023b), providing a principled methodology for imputing missing potential outcomes under low-rank structural assumptions. This fusion of causal inference and matrix completion is particularly pertinent in contemporary applications, where treatment regimes are multi-faceted (e.g., varying exposure levels in online advertising or policy intensities) and data are *missing not at random* (MNAR) (Schnabel et al., 2016; Ma & Chen, 2019), as the very process of treatment assignment is often driven by unobserved latent factors.

Within this broader context, we focus on the problem of *causal matrix completion under multiple, discrete treatment levels*. This extends the binary treatment framework introduced by Agarwal et al. (2023b) to accommodate a more realistic and complex setting. The problem shifts from the task of completing a single matrix of potential outcomes to addressing a collection of matrices (or equivalently, a three-dimensional tensor), one for each treatment level, where only one outcome per unit is observed. While previous work, such as the *Synthetic Interventions* framework (Agarwal et al., 2020), also considers multi-level treatments, it is primarily designed for panel data with a temporal dimension, aiming to estimate average effects over post-treatment periods. In contrast, our approach abstractly separates the temporal aspect, focusing on *entry-wise* counterfactual estimation in a general matrix completion setting under MNAR. This distinction is crucial: our goal is the fine-grained identification of causal effects at the unit level, rather than aggregate temporal effects. The theoretical complexity of this multi-treatment setting, particularly under MNAR conditions, remains largely unexplored, presenting formidable identification and estimation challenges. (Cinelli et al., 2025)

A significant challenge arises when treatment assignments are highly imbalanced—some treatment levels are *data-scarce*. (Cinelli et al., 2025) Existing methods, such as the Synthetic Nearest Neighbors (SNN) algorithm (Agarwal et al., 2023b), face difficulties in this regime: a direct extension requires anchor rows and columns exclusively from the target treatment, which may not exist under sparsity. Other nearest-neighbor or matrix-completion approaches

---

[1]School of Statistics and Data Science, Shanghai University of Finance and Economics, Shanghai, China [2]Institute of Big Data Research, Shanghai University of Finance and Economics, Shanghai, China [3]Institute for Interdisciplinary Information Sciences, Tsinghua University, Beijing, China. Correspondence to: Zhiheng Zhang <zhangzhiheng@mail.shufe.edu.cn>.

*Proceedings of the 43ʳᵈ International Conference on Machine Learning*, Seoul, South Korea. PMLR 306, 2026. Copyright 2026 by the author(s).

[1]Our code is publicly available at this link.

for multi-treatment MNAR data (e.g., CY (Choi & Yuan, 2024), TS-NN (Sadhukhan et al., 2025), KernelNN (Choi et al., 2025), DRNN (Dwivedi et al., 2024)) alleviate some challenges but do not exploit cross-treatment invariance of local imputation coefficients required for counterfactual estimation. This issue is not merely a data limitation; it reflects a fundamental *inefficiency* in the utilization of available information. The key insight bridging this gap is that, under the assumption of shared latent row factors across treatments, the imputation coefficients can be *identified* by leveraging data from *multiple* treatment levels. Previous multi-treatment frameworks, including Synthetic Interventions, either fail to exploit this cross-treatment identifiability for entry-wise estimation or are not designed to tackle the general MNAR matrix completion problem that we address.

To overcome these challenges, we propose the **M**ixed **S**ynthetic **N**earest **N**eighbors (**MSNN**) algorithm. Our method systematically tackles the data scarcity problem by relaxing the stringent same-treatment requirement inherent in the SNN approach. The key innovation lies in the introduction of *Mixed Anchor Rows (MAR)* and *Mixed Anchor Columns (MAC)*, enabling the imputation coefficient to be estimated from a block of data that spans *multiple treatment levels*, while preserving the target row's data from the treatment level of interest. This is made possible by the shared latent row factor assumption, which ensures the identifiability of the imputation coefficient across treatments. The core technical challenge is managing the heterogeneous scales and variances introduced by mixing treatments, which we address through the careful design of appropriate weights.

The most counterintuitive and theoretically impactful result is the *exponential improvement in sample efficiency* for sparse treatment levels. We demonstrate that, under a Missing Completely at Random (MCAR) treatment assignment, the expected number of usable data subgroups for MSNN, $\mathbb{E}[K_{\text{MSNN}}]$, surpasses that of SNN, $\mathbb{E}[K_{\text{SNN}}]$, by a factor of $\left[\sum_{d'} (p_{d'}/p_d)^{r+1}\right]^c$, where $p_d$ is the observation probability for treatment $d$, and $r, c$ are the sizes of the anchor sets. This exponential improvement significantly enhances the feasibility of causal estimation in data-scarce environments. Importantly, MSNN retains the finite-sample error bounds and asymptotic normality of the original SNN estimator, ensuring that this efficiency gain does not come at the cost of statistical rigor. Similar exponential improvement also holds under Missing Not at Random (MNAR). This challenges the conventional wisdom that estimating effects for a rare treatment necessarily requires more data from that specific treatment. Instead, we show that information from more prevalent treatments can be effectively leveraged to learn about rare treatments via shared latent structures. Our contributions are summarized as follows:

1. We formalize the problem of entry-wise causal matrix completion under multiple MNAR treatment levels and establish a novel identification result, demonstrating that imputation coefficients can be shared across treatments under a shared latent row factor assumption.

2. We propose the MSNN algorithm, which integrates data across treatment levels via Mixed Anchor Sets. We prove that MSNN retains the desirable statistical properties (finite-sample bound, asymptotic normality) of SNN, while achieving exponential improvements in sample efficiency for sparse treatments.

3. Through simulations and a case study on California's tobacco control policy, we demonstrate that MSNN reliably estimates effects for data-scarce treatments where SNN fails, highlighting its practical applicability.

## 2. Related Work

Our work generalizes MNAR matrix completion to multiple treatments via tensor completion, extending the binary case (Agarwal et al., 2023b) and differing from time-average estimation (Agarwal et al., 2020), time-sequential assumptions (Gao et al., 2025), or independent missingness (Zhang et al., 2025a). Nearest-neighbor methods for multiple treatments include (Choi & Yuan, 2024; Sadhukhan et al., 2025; Choi et al., 2025; Dwivedi et al., 2024). Unlike counterfactual regression that assumes i.i.d. samples (Johansson et al., 2016; Shalit et al., 2017), we handle structured missingness via joint low-rank learning and balancing. Analogous to causal recommendation that treats exposure as treatment (Bonner & Vasile, 2018; Schnabel et al., 2016; Wang et al., 2021; Sato et al., 2020), our Mixed Anchor Blocks perform debiased completion. A more detailed discussion is provided in Appendix A.1.

## 3. Problem Setup and Identification

This section formalizes the problem and introduces necessary notations and assumptions, then gives the identification result.

We define the exposure levels: $\mathcal{L} = \{1, 2, \cdots, l\}$ and treatment assignment $\boldsymbol{D}$: for consumer $i \in [m]$ and product $j \in [n]$, the assigned exposure level is $D_{ij} \in \{0\} \cup \mathcal{L}$, where $D_{ij} = 0$ denotes no exposure. Potential outcomes are defined as $\boldsymbol{Y}$: for consumer $i$ and product $j$, under exposure level $d$ ($d \in \mathcal{L}$), the platform obtains a benefit $Y_{ij}^{(d)} \in \mathbb{R}$. The expected potential outcome is $A_{ij}^{(d)}$, which represents a latent signal matrix assumed to be low-rank across users and items. Specifically, suppose the rank is $r$, then $\boldsymbol{A}^{(d)} = \boldsymbol{U}^{(d)} \boldsymbol{V}^{(d)^\top}$, where $\boldsymbol{U}^{(d)} \in \mathbb{R}^{m \times r}$, $\boldsymbol{V}^{(d)} \in \mathbb{R}^{n \times r}$. The zero-mean noise is defined as $\epsilon_{ij}^{(d)} = Y_{ij}^{(d)} - A_{ij}^{(d)}$. We consider general MNAR regime: $\boldsymbol{D} \perp\!\!\!\perp \boldsymbol{Y} \mid \boldsymbol{A}$, meaning

that the assignment mechanism depends on the latent attributes but not on the random noise. Only the observed outcome $\tilde{Y}$ under the assigned treatment $D$ is observed:
$\tilde{Y}_{ij} = \begin{cases} Y_{ij}^{(D_{ij})} & , D_{ij} \geq 1 \\ * & , D_{ij} = 0 \end{cases}$. Here $*$ means this entry is non-observable (missing).

**Problem Statement.** Given noisy observation $\tilde{Y}$ with the treatment assignment $\mathbf{D}$, under causal model $D \perp\!\!\!\perp Y \mid A$, we are tasked to find an estimator of the latent expected potential outcome $A$ from $\tilde{Y}$, and provide an entry-wise upper bound of error of estimation and ground truth: $\left\| \hat{A} - A \right\|_\infty$.

### 3.1. Basic Assumptions

Below, we introduce basic assumptions for our low-rank multiple matrix completion model.

**Assumption 3.1** (Low-rank factorization per layer)**.** For each $(i, j, d) \in [m] \times [n] \times \mathcal{L}$, $Y_{ij}^{(d)} = \left\langle u_i^{(d)}, v_j^{(d)} \right\rangle + \epsilon_{ij}^{(d)}$,

where $u_i^{(d)}, v_j^{(d)} \in \mathbb{R}^r$ are latent factors specific to treatment level $d$. Equivalently, $\boldsymbol{Y}^{(d)} = \boldsymbol{U}^{(d)} \boldsymbol{V}^{(d)\top} + \boldsymbol{E}^{(d)}$, where $u_i^{(d)}, v_j^{(d)}$ are the $i^{th}$ and $j^{th}$ rows of $\boldsymbol{U}^{(d)} \in \mathbb{R}^{m \times r}$, $\boldsymbol{V}^{(d)} \in \mathbb{R}^{n \times r}$ respectively.

Below we denote $\boldsymbol{U}, \boldsymbol{V}$ and $\boldsymbol{E}$ as the collection of $\boldsymbol{U}^{(d)}, \boldsymbol{V}^{(d)}, \boldsymbol{E}^{(d)}$ over treatment $d$.

**Assumption 3.2** (Selection on latent factors)**.** For any treatment $\boldsymbol{D}$, $\mathbb{E}\left[\boldsymbol{E}|\boldsymbol{U}, \boldsymbol{V}, \boldsymbol{D}\right] = 0$.

*Remark* 3.3. By taking conditional expectation over $\boldsymbol{D}$, $\mathbb{E}\left[\boldsymbol{E}|\boldsymbol{U}, \boldsymbol{V}\right] = \mathbb{E}_{\boldsymbol{D}}\left[\mathbb{E}\left[\boldsymbol{E}|\boldsymbol{U}, \boldsymbol{V}, \boldsymbol{D}\right]|\boldsymbol{U}, \boldsymbol{V}\right] = 0$, further implying $\mathbb{E}\left[\boldsymbol{Y}|\boldsymbol{U}, \boldsymbol{V}, \boldsymbol{D}\right] = \mathbb{E}\left[\boldsymbol{Y}|\boldsymbol{U}, \boldsymbol{V}\right]$, which is a conditional mean independence.

**Assumption 3.4** (Linear span inclusion on latent row factors)**.** There exists a universal constant $\mu \in \mathbb{N}$, such that given arbitrary $(i, d) \in [m] \times \mathcal{L}$, for all $\mathcal{I}^{(d)}(i) \subseteq [m] \backslash \{i\}$ satisfying $\left| \mathcal{I}^{(d)}(i) \right| \geq \mu$, $u_i^{(d)}$ is a linear combination of $u_{l \in \mathcal{I}^{(d)}(i)}^{(d)}$. To rephrase, there exists a $\beta^{(d)}\left(\mathcal{I}^{(d)}(i)\right) \in \mathbb{R}^{\mathcal{I}^{(d)}(i)}$ such that $u_i^{(d)} = \sum_{l \in \mathcal{I}^{(d)}(i)} \beta_l^{(d)}\left(\mathcal{I}^{(d)}(i)\right) u_l^{(d)}$.

The above three assumptions simply extend the basic assumptions of the binary causal matrix completion framework (Agarwal et al., 2023b) to the multiple treatment setting without considering the connection among different treatment levels. We characterize this low-rank structure across different treatments in the following assumption:

**Assumption 3.5** (Shared latent row factors)**.** All the treatment levels $d \in \mathcal{L}$ share the same latent row factors, $u_i^{(d)} \equiv u_i$.

**Discussion on Assumption 3.5**. This assumption is necessary for enabling cross-treatment data integration in our

MSNN: row-mode latent factors are shared across treatment levels, while treatment heterogeneity is captured by treatment-specific loading matrices. This shared structure allows MSNN to borrow information from data-rich treatments for estimating data-scarce treatments. Appendix A.2 provides further interpretation, examples, relation to prior low-rank causal models, and robustness analysis under moderate violations.

### 3.2. Identification of Multiple Treatment Levels

Below we define $A := \mathbb{E}\left[\boldsymbol{Y}|\boldsymbol{U}, \boldsymbol{V}\right]$ as the ground truth. The theoretical guarantee of Mixed Synthetic Nearest Neighbors (MSNN) we introduce below is the treatment-invariance of the coefficient $\beta$, which is followed by the low-rank assumption on treatment represented by latent row factor (Assumption 3.5).

**Lemma 3.6.** *Under Assumption 3.4 and 3.5, the index set $\mathcal{I}^{(d)}(i)$ and coefficient $\beta^{(d)}\left(\mathcal{I}^{(d)}(i)\right)$ are irrelevant to treatment $d$, i.e. $\forall d' \in \mathcal{L}$, $u_i^{(d')} = \sum_{l \in \mathcal{I}^{(d)}(i)} \beta_l^{(d)}\left(\mathcal{I}^{(d)}(i)\right) u_l^{(d')}$ for any valid $\mathcal{I}^{(d)}(i)$ and $\beta_l^{(d)}\left(\mathcal{I}^{(d)}(i)\right)$.*

Proof is presented in Appendix D.1. Due to this lemma, the index set and coefficient are shared across treatments, and thus we omit the index $(d)$ for $\mathcal{I}^{(d)}(i)$ and $\beta^{(d)}$ in the derivation below.

**Theorem 3.7.** *Under Assumptions 3.1, 3.2, 3.4, 3.5, given $(i, j, d)$ and $\mathcal{I}(i) \subseteq [m] \backslash \{i\}$ such that $D_{lj, l \in \mathcal{I}(i)} = d$, then*

$$A_{ij}^{(d)} = \sum_{l \in \mathcal{I}(i)} \beta_l\left(\mathcal{I}(i)\right) \mathbb{E}\left[\tilde{Y}_{lj} \Big| \boldsymbol{U}, \boldsymbol{V}, \boldsymbol{D}\right]. \quad (1)$$

Proof is presented in Appendix D.1. Similar to the identification theorem in Agarwal et al. (2023b), one can estimate the potential outcome $A_{ij}^{(d)}$ through the linear model parameter $\beta$. The main difference under multiple treatment is that, here the estimation of a valid $\beta$ does not require the data from the same treatment level.

## 4. From SNN to MSNN: Data Integration across Treatment Levels

This section introduces MSNN (Mixed Synthetic Nearest Neighbors) to address the lack of data under multiple treatments.

### 4.1. SNN under Multiple Treatments

Denote $x(d) := \left[\tilde{Y}_{aj}\right]_{a \in \mathrm{AR}(d)}$, $q(d) := \left[\tilde{Y}_{ib}\right]_{b \in \mathrm{AC}(d)}$. ($q$ is a row vector. ) From the definition of $\mathrm{AR}(d)$, $x(d) = \left[Y_{aj}^{(d)}\right]_{a \in \mathrm{AR}(d)}$, $q(d) = \left[Y_{ib}^{(d)}\right]_{b \in \mathrm{AC}(d)}$, which is exactly

**Algorithm 1** SNN$(i,j,d)$

Input: $\left\{\lambda^{(k)}(d) : k \in [K_{\text{SNN}}], d \in \mathcal{L}\right\}$, $\left\{\left(\text{AR}^{(k)}(d), \text{AC}^{(k)}(d)\right) : k \in [K_{\text{SNN}}]\right\}$ with mutually disjoint sets $\left\{\text{AR}^{(k)}(d) : k \in [K_{\text{SNN}}]\right\}$.

**for** $k \in [K_{\text{SNN}}]$ **do**

   1. Define $\boldsymbol{S}^{(k)}(d) = \left[\tilde{Y}_{ab} : (a,b) \in \text{AR}^{(k)}(d) \times \text{AC}^{(k)}(d)\right]$

   2. Compute SVD decomposition $\boldsymbol{S}^{(k)}(d) \leftarrow \sum_{l \geq 1} \hat{\tau}_l^{(k)}(d) \hat{u}_l^{(k)}(d) \otimes \hat{v}_l^{(k)}(d)$

   3. Compute $\hat{\beta}^{(k)}(d) \leftarrow \left(\sum_{l \leq \lambda^{(k)}(d)} \left(1/\hat{\tau}_l^{(k)}(d)\right) \hat{u}_l^{(k)}(d) \otimes \hat{v}_l^{(k)}(d)\right) q^{(k)\top}(d)$

   4. Compute $\hat{A}_{ij,k}(d) \leftarrow \left\langle x^{(k)}(d), \hat{\beta}^{(k)}(d)\right\rangle$

**end for**

Output $\hat{A}_{ij}^{(d)} \leftarrow \frac{1}{K_{\text{SNN}}} \sum_{k=1}^{K_{\text{SNN}}} \hat{A}_{ij,k}(d)$

---

**Algorithm 2** MSNN$(i,j,d)$

Input: $\left\{\lambda^{(k)}(d) : k \in [K_{\text{MSNN}}], d \in \mathcal{L}\right\}$, $\left\{\left(\text{MAR}^{(k)}(d), \text{MAC}^{(k)}(d)\right) : k \in [K_{\text{MSNN}}]\right\}$ with mutually disjoint sets $\left\{\text{MAR}^{(k)}(d) : k \in [K_{\text{MSNN}}]\right\}$ w.r.t. column treatment levels $d\left(b : b \in \text{MAC}^{(k)}(d)\right)$.

**for** $k \in [K_{\text{MSNN}}]$ **do**

   1. Define $\boldsymbol{S}_w^{(k)}(d) = \left[w(b, d(b)) \cdot \tilde{Y}_{ab} : (a,b) \in \text{MAR}^{(k)}(d) \times \text{MAC}^{(k)}(d)\right]$

   2. Compute SVD decomposition $\boldsymbol{S}_w^{(k)}(d) \leftarrow \sum_{l \geq 1} \hat{\tau}_l^{(k)}(d) \hat{u}_l^{(k)}(d) \otimes \hat{v}_l^{(k)}(d)$

   3. Compute $\hat{\beta}^{(k)}(d) \leftarrow \left(\sum_{l \leq \lambda^{(k)}(d)} \left(1/\hat{\tau}_l^{(k)}(d)\right) \hat{u}_l^{(k)}(d) \otimes \hat{v}_l^{(k)}(d)\right) q_w^{(k)\top}(d)$

   4. Compute $\hat{A}_{ij,k}(d) \leftarrow \left\langle x^{(k)}(d), \hat{\beta}^{(k)}(d)\right\rangle$

**end for**

Output $\hat{A}_{ij}^{(d)} \leftarrow \frac{1}{K_{\text{MSNN}}} \sum_{k=1}^{K_{\text{MSNN}}} \hat{A}_{ij,k}(d)$

---

the potential outcome at treatment level $d$. $(\cdot)^{(k)}$ means the $k^{th}$ group out of total group $K_{\text{SNN}}$. Now we present the SNN algorithm in Algorithm 1.

To estimate potential outcome on $(i,j)$ under treatment $d$, trivial SNN on multiple treatment needs to select Anchor Rows (AR) and Anchor Columns (AC) specific to treatment level $d$ such that: $D_{ab:a \in \text{AR}(d), b \in \text{AC}(d)} = d$, $D_{aj:a \in \text{AR}(d)} = d$, $D_{ib:b \in \text{AC}(d)} = d$.

### 4.2. Mixed Synthetic Nearest Neighbors: Data Combination

MSNN leverages data across different treatment levels as ensured by Theorem 3.7. The key insight is that, while the linear combination factors $x(d)$ come from the same treatment level as the estimated target treatment level $d$, the estimation of the coefficient vector $\beta$ can incorporate information from other treatments. This observation motivates relaxing the notions of Anchor Rows (AR) and Anchor Columns (AC) to their mixed counterparts - Mixed Anchor Rows (MAR) and Mixed Anchor Columns (MAC) - defined as follows.

Mixed Anchor Rows (MAR) and Mixed Anchor Columns (MAC) at treatment level $d$ for entry $(i,j)$ satisfy the following conditions: 1. $\forall a \in \text{MAR}(d), b \in \text{MAC}(d)$, $D_{ab:a \in \text{MAR}(d), b \in \text{MAC}(d)} = D_{ib:b \in \text{MAC}(d)} =: d(b) \neq 0$ (where we call column treatment level below); 2. $\forall a \in \text{MAR}(d)$, $D_{aj:a \in \text{MAR}(d)} = d$.

Intuitively, this means that within the block $\text{MAR}(d) \times \{\text{MAC}(d) \cup \{j\}\}$, every column shares the same (non-zero) treatment assignment as the row $i$ corresponding to those columns. Now $x(d) = \left[Y_{aj}^{(d)}\right]_{a \in \text{MAR}(d)}$ remains, however

$q(d) = \left[Y_{ib}^{(d(b))}\right]_{b \in \text{MAC}(d)}$ where entries may consist of different treatment levels.

To balance the probable scale and variance heterogeneity introduced by mixed treatment levels, we also introduce positive weight functions $w(j,d) : [n] \times \mathcal{L} \to \mathbb{R}_+$.

We denote the weighted $q(d)$ as $q_w(d) := \left[w(b, D_{ib}) \cdot \tilde{Y}_{ib}\right]_{b \in \text{MAC}(d)} = \left[w(b, d(b)) \cdot Y_{ib}^{(d(b))}\right]_{b \in \text{MAC}(d)}$. Now we introduce our MSNN algorithm in Algorithm 2 above. An illustrative comparison between SNN and MSNN is also provided in Figure 1.

*Remark* 4.1 (Utility of weight $w(b, d(b))$). Since $\boldsymbol{S}_w^{(k)}(d)$ pools observations from different treatment levels, scale heterogeneity across treatments may make the anchor block ill-conditioned and destabilize the SVD step in Algorithm 2. The weight $w(b, d(b))$ is introduced to normalize columns from different treatment scales. For example, an unweighted block $\text{diag}(1, C)$ has condition number $C \gg 1$, while weighting the second column by $1/C$ gives the identity matrix and condition number 1. Therefore, a natural choice is $w(b, d(b)) = 1/f(d(b))$, which brings different treatment levels to a comparable scale. When $f(d)$ is unknown, it can be replaced by an estimated scale, such as the maximum observed absolute entry within each treatment. We revisit the selection of $w(b, d(b))$ and see how it affects the estimation error in Remark 5.7.

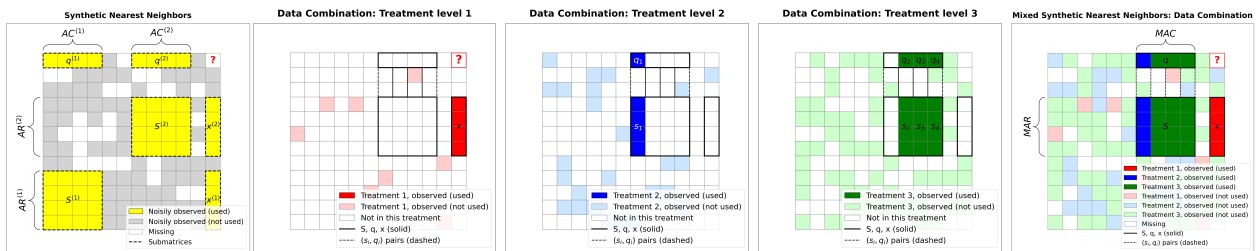

*Figure 1.* Comparison between SNN and MSNN. The leftmost subfigure illustrates the SNN algorithm with $K_{\text{SNN}} = 2$: it requires $\boldsymbol{S}^{(k)}$, $q^{(k)}$ and $x^{(k)}$ are all fully observed at treatment level as the same of the estimated treatment, which is rare under data-scarce levels (e.g. the "red" level in the second subfigure). The rest four subfigures explain the procedure of MSNN for a specific subgroup $k$: given entry $(i, j)$ and estimated treatment level (here is "red"), one needs to find a fully observed $x^{(k)}$ under same "red" level, but the $\boldsymbol{S}^{(k)}$ and $q^{(k)}$ can be integrated from other treatments (here: "blue" and "green" level). The only requirement is that for each column of $\boldsymbol{S}^{(k)}$ (namely, $s_i^{(k)}$), its treatments should be the same as the treatment of corresponding $q_i^{(k)}$, see the third and fourth subfigures.

---

**Algorithm 3** MixedAnchorSubMatrix $(i, j, d)$

---

Input: `createGraph, maxBiclique`
1. Assign $\boldsymbol{B}^{(i,j,d)} = [\mathbf{1}\{D_{ab} = D_{ib} \neq 0, D_{aj} = d, a \neq i, b \neq j\}]_{a \in [m], b \in [n]}$
2. Generate $\mathcal{G} \leftarrow$ `createGraph`$(\boldsymbol{B}^{(i,j,d)})$
3. Compute $(\mathcal{V}_1', \mathcal{V}_2', \mathcal{E}') \leftarrow$ `maxBiclique`$(\mathcal{G})$
4. Output $\text{MAR}(d) \leftarrow \mathcal{V}_1', \text{MAC}(d) \leftarrow \mathcal{V}_2'$

---

### 4.3. Finding Mixed Anchor Rows and Columns

Given a target entry $(i, j, d)$ and a treatment assignment $\boldsymbol{D}$, we can construct the $\{0, 1\}$ valued matrix $\boldsymbol{B}^{(i,j,d)} \in \{0, 1\}^{m \times n}$ by $\boldsymbol{B}^{(i,j,d)} = [\mathbf{1}\{D_{ab} = D_{ib} \neq 0, D_{aj} = d, a \neq i, b \neq j\}]_{a \in [m], b \in [n]}$, which indicates whether an entry of $\tilde{Y}$ is usable for constructing $\text{MAR}(d)$ and $\text{MAC}(d)$. Given $K_{\text{MSNN}}$ subgroups of $(\text{MAR}^{(k)}(d), \text{MAC}^{(k)}(d))$, equivalently they represent $K_{\text{MSNN}}$ all-1 submatrices of $\boldsymbol{B}^{(i,j,d)}$ with disjoint rows.

Now if given $|\text{MAR}(d)| = r$ and $|\text{MAC}(d)| = c$, the problem equivalently reduces to selecting $K_{\text{MSNN}}$ all-1 submatrices from $\boldsymbol{B}^{(i,j,d)}$ with disjoint rows. By modifying algorithm `AnchorSubMatrix` discussed in Agarwal et al. (2023b), we can construct $\text{MAR}(d)$ and $\text{MAC}(d)$ by our Algorithm 3: `MixedAnchorSubMatrix`. Then given $\text{MAR}(d)$ and $\text{MAC}(d)$, we construct $K_{\text{MSNN}}$ subgroups by selecting $\text{MAC}^{(k)}(d) = \text{MAC}(d)$, while partition $\text{MAR}(d)$ into $K_{\text{MSNN}}$ subgroups of equal size randomly.

Here we assume `createGraph` and `maxBiclique` as two known algorithms. `createGraph` takes input $\boldsymbol{B}^{(i,j,d)}$ as a bipartite incident matrix for a bipartite graph $\mathcal{G}$ (that is, $\mathcal{G} = (\mathcal{V}_1, \mathcal{V}_2, \mathcal{E})$ where $|\mathcal{V}_1| = m$ and $\mathcal{V}_2 = n$; for $v_a \in \mathcal{V}_1$ and $v_b \in \mathcal{V}_2$, $(v_a, v_b) \in \mathcal{E}$ if and only if $\boldsymbol{B}_{ab}^{(i,j,d)} = 1$) and outputs $\mathcal{G}$. `maxBiclique` takes bipartite $\mathcal{G}$ as input and then outputs a maximal bipartite clique.

## 5. Theoretical Results

We further add Assumptions to establish results on finite-sample bound and asymptotic normality.

### 5.1. Additional Assumptions

By following the notation in Agarwal et al. (2023b), below we denote $\mathcal{E} = \{\boldsymbol{U}, \boldsymbol{V}, \boldsymbol{D}\}$. Following the requirements of SNN (Agarwal et al., 2023b), we further state the following assumptions under the multiple treatment setting:

**Assumption 5.1** (Bounded expected potential outcomes with treatment heterogeneity). Conditioned on $\mathcal{E}$, $A_{ij}^{(d)} \in [-f(d), f(d)]$, where $f(d) : \mathcal{L} \to \mathbb{R}^+$ characterizes the range of potential outcome for each treatment.

**Assumption 5.2** (Sub-gaussian noise with treatment heterogeneity). Conditioned on $\mathcal{E}$, the error $\epsilon_{ij}^{(d)}$ are independent sub-gaussian mean-zero random variables with $\mathbb{E}\left[\left(\epsilon_{ij}^{(d)}\right)^2\right] = \left(\sigma_{ij}^{(d)}\right)^2$ and $\left\|\epsilon_{ij}^{(d)}\right\|_{\psi_2} \leq C\sigma_{ij}^{(d)}$, where $\sigma_{ij}^{(d)} \leq f(d)\sigma$ for some universal $\sigma > 0$ and constant $C > 0$.

**Assumption 5.3** (Well balanced spectra under appropriate scaling). For every treatment level $d \in \mathcal{L}$, conditioned on $\mathcal{E}$ and given $(i, j)$ with subgroup $k$, if $w(b, d(b)) = 1/f(d(b))$ ($b \in \text{MAC}^{(k)}(d)$), the non-zero singular values of $\mathbb{E}\left[\boldsymbol{S}_w^{(k)}(d)\big|\mathcal{E}\right]$ (namely, $\tau_{l \in [r^{(k)}]}^{(k)}(d)$) are well balanced, i.e. there exists universal constants $c, c' > 0$ such that $\tau_{r^{(k)}}^{(k)}(d)/\tau_1^{(k)}(d) \geq c$ and $\tau_1^{(k)}(d) \leq c' \cdot \left|\text{MAR}^{(k)}(d)\right|\left|\text{MAC}^{(k)}(d)\right|$.

**Assumption 5.4** (Subspace inclusion). Conditioned on $\mathcal{E}$, given $(i, j, d)$ with subgroup $k$,

$$\mathbb{E}\left[x^{(k)}(d)\big|\boldsymbol{U}, \boldsymbol{V}, \boldsymbol{D}\right] \in \text{Col}\left(\mathbb{E}\left[\boldsymbol{S}_w^{(k)}(d)\big|\boldsymbol{U}, \boldsymbol{V}, \boldsymbol{D}\right]\right).$$

## 5.2. Preservation of Finite-sample Bound and Asymptotic Normality

Under the above assumptions, the data that MSNN leverages (in Algorithm 2) exactly satisfy the conditions that SNN requires, thus similar conclusions of finite-sample bound and asymptotic normality (refer to Theorem 2 and 3 of Agarwal et al. (2023b)) can be transferred from SNN to MSNN:

**Theorem 5.5** (Finite-sample error bound). *Conditioned on $\mathcal{E}$, for a given entry $(i,j)$ and treatment level $d$, suppose $\left|\mathrm{MAR}^{(k)}(d)\right| \geq \mu$ for all subgroup $k \in [K_{\mathrm{MSNN}}]$, let Assumptions 3.1-3.5 and 5.1-5.4 hold. Further let $K_{\mathrm{MSNN}} = o\left(\min_k \left|\mathrm{MAC}^{(k)}\right|^{10} \left|\mathrm{MAR}^{(k)}\right|^{10}\right)$. Finally set $\lambda^{(k)} = \mathrm{rank}\left(\mathbb{E}\left[\boldsymbol{S}_w^{(k)}\right]\right)$, $w(b, d(b)) = 1/f(d(b))$, then $\hat{A}_{ij}^{(d)} - A_{ij}^{(d)} = f(d) \cdot O_p\left(\frac{1}{K_{\mathrm{MSNN}}}\left\{error + \left[\sum_{k=1}^{K_{\mathrm{MSNN}}} \|\tilde{\beta}^{(k)}\|_2^2\right]^{1/2}\right\}\right)$,*

*where $error := \sum_{k=1}^{K_{\mathrm{MSNN}}} (error_{k,1} + error_{k,2})$. Here, $\tilde{\beta}^{(k)} := \mathcal{P}_{\mathbb{E}\left[\boldsymbol{S}_w^{(k)}(d)\big|\mathcal{E}\right]} \beta^{(k)}$ is the projection of $\beta^{(k)}$ onto the column space of $\mathbb{E}\left[\boldsymbol{S}_w^{(k)}(d)\big|\mathcal{E}\right]$, and*

$$error_{k,1} := \frac{(r^{(k)})^{1/2}}{|\mathrm{MAC}^{(k)}(d)|^{1/4}},$$

$$error_{k,2} := \frac{(r^{(k)})^{3/2}\|\tilde{\beta}^{(k)}\|_1 \log^{1/2}(|\mathrm{MAC}^{(k)}(d)||\mathrm{MAR}^{(k)}(d)|)}{\min\{|\mathrm{MAC}^{(k)}(d)|^{1/2}, |\mathrm{MAR}^{(k)}(d)|^{1/2}\}}.$$

**Theorem 5.6** (Asymptotic Normality). *For a given entry $(i,j)$ and treatment level $d$, let the setup of Theorem 5.5 hold. Define $(\tilde{\sigma}^{(k)}(d))^2 := \sum_{l \in \mathrm{MAR}^{(k)}(d)} (\tilde{\beta}_l^{(k)} \sigma_{lj}^{(d)})^2$. Further assume that*

*(i) $K_{\mathrm{MSNN}} \to \infty$,*

*(ii) $|\mathrm{MAC}^{(k)}(d)|, |\mathrm{MAR}^{(k)}(d)| \to +\infty$ for each $k$,*

*(iii) For each $k$, $r^{(k)}\|\tilde{\beta}^{(k)}\|_1^2 \log(|\mathrm{MAC}^{(k)}(d)||\mathrm{MAR}^{(k)}(d)|) = o\left(\min\{|\mathrm{MAC}^{(k)}(d)|, |\mathrm{MAR}^{(k)}(d)|\}\right)$,*

*(iv) $f(d) \cdot error = o\left(\left[\sum_{k=1}^{K_{\mathrm{MSNN}}} (\tilde{\sigma}^{(k)}(d))^2\right]^{1/2}\right).$*

*Then conditioned on $\mathcal{E}$, $\frac{K_{\mathrm{MSNN}}(\hat{A}_{ij}^{(d)} - A_{ij}^{(d)})}{\left[\sum_{k=1}^{K_{\mathrm{MSNN}}} (\tilde{\sigma}^{(k)}(d))^2\right]^{1/2}} \xrightarrow{d} \mathcal{N}(0, 1).$*

Proofs of Theorem 5.5 and 5.6 are provided in Appendix D.2, followed from reductions to the original SNN analysis. These two Theorems have the same form as the finite-sample bound and asymptotic normality of SNN, by replacing $|\mathrm{MAR}|, |\mathrm{MAC}|$ with $|\mathrm{AR}|, |\mathrm{AC}|$. Interpretation of conditioning on $\mathcal{E}$ is discussed in Remark A.2.

*Remark* 5.7 (Estimated treatment-scale weights). Theorem 5.5 and 5.6 are stated for $w(b, d(b)) = 1/f(d(b))$. Suppose

instead that the treatment scales are unknown and we use estimated positive scales $\hat{f}(d)$, i.e., $w(b, d(b)) = 1/\hat{f}(d(b))$ in estimation. Assume $f(d)/\hat{f}(d)$ is uniformly bounded by some constants, then the finite-sample bound in Theorem 5.5 remains valid with the leading scale factor $f(d)$ replaced by $\hat{f}(d) \cdot \max_{d' \in \mathcal{L}}(f(d')/\hat{f}(d'))$, while the asymptotic normality in Theorem 5.6 remains unchanged. A detailed explanation is provided in Remark D.2.

## 5.3. Sample Efficiency for Large Matrices

To exhibit the efficacy of MSNN, we illustrate that our new Algorithm 2 surpasses Algorithm 1 under the following regime: given the distribution of treatment assignment $\boldsymbol{D}$, we compare the expected number of subgroups we can extract at most from observed outcome $\tilde{\boldsymbol{Y}}$ for MSNN and SNN: $\mathbb{E}[K_{\mathrm{MSNN}}]$ and $\mathbb{E}[K_{\mathrm{SNN}}]$. We expect that under data-scarce treatment levels, $\mathbb{E}[K_{\mathrm{MSNN}}] \gg \mathbb{E}[K_{\mathrm{SNN}}]$. The proofs of sample-efficiency improvement (MCAR and MNAR) form the main novel contribution of this paper.

We analyze $\mathbb{E}\left[K_{(\cdot)}\right]$ mainly because of the following reasons from both theoretical and empirical perspectives:

1. Based on previous section, under the conditions of Theorem 5.6 (Asymptotic Normality), the error term is of order $K_{(\cdot)}^{-1/2}$. For large matrices of size $m, n \gg 1$, this section now presents theoretical guarantee that the estimation error is smaller at the rate of order $\sqrt{\frac{\mathbb{E}[K_{\mathrm{SNN}}]}{\mathbb{E}[K_{\mathrm{MSNN}}]}}$.

2. It is computationally hard to find the maximum number of all-1 submatrix from a $0-1$ valued matrix (Algorithm 3 addresses this by providing $K_{(\cdot)}$ feasible submatrices without optimal guarantee, where $K_{(\cdot)}$ is a tunable hyperparameter), especially when the shape of submatrices are large. Even for Algorithm 3, it is computationally almost unaffordable to find when $|\mathrm{MAR}|, |\mathrm{MAC}|$ exceed 10. This motivates us to consider another important problem: can we find a feasible Anchor Matrix to estimate under data-scarce treatment levels subject to the identifiability requirement that the (Mixed) Anchor Rows/Columns are no smaller than a threshold, this probability is linked to the expectation of the subgroup count:

$$\mathbb{E}\left[K_{(\cdot)}\right] = \sum_{k \geq 1} \mathbb{P}\left(K_{(\cdot)} \geq k\right)$$

$$\implies \frac{\mathbb{E}\left[K_{(\cdot)}\right]}{\max K_{(\cdot)}} \leq \mathbb{P}\left(K_{(\cdot)} \geq 1\right) \leq \mathbb{E}\left[K_{(\cdot)}\right].$$

Generally, the improvement of $\mathbb{E}\left[K_{(\cdot)}\right]$ leads to the improvement of $\mathbb{P}\left(K_{(\cdot)}\right)$. For the simulation study in the next section, we mainly focus on the data-scarce case that SNN fails to find any Anchor Matrix while MSNN achieves feasible estimation: although $\mathbb{E}\left[K_{\mathrm{SNN}}\right] \ll 1$, we have $\mathbb{E}\left[K_{\mathrm{MSNN}}\right] \gtrsim 1$.

### 5.3.1. SAMPLE EFFICIENCY UNDER MCAR

We first consider the simplest case of MCAR. Each entry $(i, j)$ is revealed i.i.d. with the following distribution:

$$\begin{cases} \mathbb{P}(D_{ij} = d) = p_d, \ \forall d \in \mathcal{L}, \\ \mathbb{P}(D_{ij} = 0) = 1 - \sum_{d' \in \mathcal{L}} p_{d'}. \end{cases} \quad (2)$$

We require for each treatment level, $p_d > 0$, and $1 - \sum_{d' \in \mathcal{L}} p_{d'} \geq 0$. We denote the largest probability as $p_{\max} := (\max_{d'} p_{d'})$, and its treatment level $d_{\max} := \arg\max_d p_d$. For simplicity we denote $\gamma := \sum_{d' \in \mathcal{L}} \left( \frac{p_{d'}}{p_{\max}} \right)^{r+1}$. Now we calculate the expectation of $K_{(\cdot)}$ for both SNN and MSNN in the following theorem:

**Theorem 5.8.** *Expectation of number of samples.*

*Consider estimation of any entry $(i, j)$ under treatment level $d$. Suppose the treatment assignment $D$ follows missing completely at random distribution, for fixed sample size $\mathrm{AR}(d) = \mathrm{MAR}(d) = r$ and $\mathrm{AC}(d) = \mathrm{MAC}(d) = c$, if the probabilities are sufficiently small, that is for some $\alpha \in (0, 1)$ the following sparsity conditions holds:*

$$mr p_d p_{\max}^{\alpha c} = o(1), \ nc\gamma p_{\max}^{(1-\alpha)r+1+\alpha} = o(1),$$

*then the expectations of $K_{(\cdot)}$ are bounded by:*

$$\mathbb{E}\left[ K_{\mathrm{SNN}}(d) \right] \in [1 - o(1), 1] \cdot \binom{m-1}{r} \binom{n-1}{c} p_d^{rc+r+c},$$

$$\mathbb{E}\left[ K_{\mathrm{MSNN}}(d) \right] \in [1 - o(1), 1] \cdot \binom{m-1}{r} \binom{n-1}{c} \gamma^c p_d^r p_{\max}^{(r+1)c}.$$

Proof is presented in Appendix D.3.2.

*Remark* 5.9. The sparsity condition above does not conflict with $\mathbb{E}[K_{\mathrm{MSNN}}(d)] = \omega(1)$. For example, consider $r = o(m)$, $c = o(n)$, then $\mathbb{E}\left[ K_{\mathrm{MSNN}}(d) \right] = \Theta\left( \frac{m^r n^c}{r!c!} \gamma^c p_d^r p_{\max}^{(r+1)c} \right) = o\left( \frac{p_{\max}^{-\alpha c}}{r! r^r c! c^c} \right)$, which does not conflict with $\mathbb{E}[K_{\mathrm{MSNN}}(d)] = \omega(1)$ for fixed $r, c$.

Based on the above theorem, we analyze the sample efficiency of MSNN quantitatively:

**Corollary 5.10.** *Exponential sample efficiency improvement of MSNN compared to trivial SNN. Under the setting of Theorem 5.8, for estimation of entry $(i, j)$ at treatment level $d$ we have*

$$\frac{\mathbb{E}[K_{\mathrm{MSNN}}(d)]}{\mathbb{E}[K_{\mathrm{SNN}}(d)]} = (1 + o(1)) \left[ \sum_{d' \in \mathcal{L}} \left( \frac{p_{d'}}{p_d} \right)^{r+1} \right]^c.$$

**Corollary 5.11.** *Estimation efficiency compared to the treatment level with the most data. Under the setting of Theorem 5.8, for estimation of entry $(i, j)$ at treatment level $d$ compared the level with most data $d_{\max}$,*

$$\frac{\mathbb{E}[K_{\mathrm{SNN}}(d)]}{\mathbb{E}[K_{\mathrm{MSNN}}(d_{\max})]} = O\left( \gamma^{-c} \left( \frac{p_d}{p_{\max}} \right)^{rc+r+c} \right),$$

$$\frac{\mathbb{E}[K_{\mathrm{MSNN}}(d)]}{\mathbb{E}[K_{\mathrm{MSNN}}(d_{\max})]} = O\left( \left( \frac{p_d}{p_{\max}} \right)^r \right).$$

Proofs are presented in Appendix D.3.2.

*Remark* 5.12. Corollary 5.10 and 5.11 together guarantee efficient estimation under data heterogeneity. For a treatment level with sparse data, Corollary 5.10 shows that the expected number of samples $K_{\mathrm{MSNN}}$ increases by a factor of at least the power of $rc$ compared to $K_{\mathrm{SNN}}$, which is an exponential improvement, while Corollary 5.11 shows that the relative efficiency gap between the data-sparsest and the data-richest levels is efficiently reduced - specifically, its dependence on the proportion $p_d/p_{d_{\max}}$ is reduced from a quadratic order $rc$ to a linear order $r$.

### 5.3.2. SAMPLE EFFICIENCY UNDER MNAR

After the discussion of MCAR, we generalize the exponential improvement of sample efficiency above to general MNAR. By generalizing the standard logistic function $\sigma(x) = \frac{e^x}{1+e^x}$ discussed by Ma & Chen (2019), we consider the following MNAR setting: each entry at $(i, j)$ is revealed with $\mathbb{P}(D_{ij} = d) = p_{\mathrm{MNAR}}\left( A_{ij}^{d' \in \mathcal{L}}, d \right)$, where $p_{\mathrm{MNAR}}$ is a multi-variant softmax function defined by

$$p_{\mathrm{MNAR}}\left( A_{ij}^{d' \in \mathcal{L}}, d \right) = \frac{e^{\lambda_d A_{ij}^d + \delta_d}}{s_0 + \sum_{d' \in \mathcal{L}} e^{\lambda_{d'} A_{ij}^{d'} + \delta_{d'}}}. \quad (3)$$

Here $\lambda_d, \delta_d \in \mathbb{R}$, $s_0 \in \mathbb{R}_+ \cup \{0\}$ are chosen hyperparameters that control the observation probability of each treatment level.

**Corollary 5.13** (Informal). *Exponential sample efficiency improvement of MSNN compared to SNN under MNAR. Consider estimation of any entry $(i, j)$ under treatment level $d$. Suppose the treatment assignment $D$ follows missing not at random distribution defined above, for fixed sample size $\mathrm{AR}(d) = \mathrm{MAR}(d) = r$ and $\mathrm{AC}(d) = \mathrm{MAC}(d) = c$, if the probabilities are sufficiently small, then the sample efficiency is lower bounded by*

$$\mathbb{E}\left[ K_{\mathrm{MSNN}}(d) \right] / \mathbb{E}\left[ K_{\mathrm{SNN}}(d) \right] \geq (1 - o(1)) \cdot (1 + \chi(d))^c,$$

*where* $\chi(d) := \left( \sum_{d' \in \mathcal{L} \setminus \{d\}} e^{(r+1)\left( -|\lambda_{d'}| f(d') + \delta_{d'} \right)} \right) \cdot$ $e^{(r+1)(-|\lambda_d| f(d) - \delta_d)}$.

The complete version of this corollary and its proofs are presented in Appendix D.3.3.

*Remark* 5.14. By setting $\lambda_d = 0$, $\delta_d = \ln p_d$, and $s_0 = 1 - \sum_{d' \in \mathcal{L}} p_{d'}$, Corollary 5.13 exactly reduces to the result of MCAR discussed in section 5.3.1.

*Table 1.* Performance comparison of MSNN and SNN on MCAR data.

| Algorithm | Low ($p(d) = 0.01$) | | Medium ($p(d) = 0.025$) | | High ($p(d) = 0.05$) | |
|---|---|---|---|---|---|---|
| | FR % ($\uparrow$) | MRE ($\downarrow$) | FR % ($\uparrow$) | MRE ($\downarrow$) | FR % ($\uparrow$) | MRE ($\downarrow$) |
| SNN | $0.02 \pm 0.02$ | $0.806 \pm 0.240$ | $1.20 \pm 0.36$ | $0.577 \pm 0.110$ | $11.34 \pm 0.82$ | $0.515 \pm 0.046$ |
| MSNN (ours) | $\mathbf{4.69 \pm 1.11}$ | $\mathbf{3.91 \pm 1.09} \times 10^{-2}$ | $\mathbf{63.73 \pm 5.67}$ | $\mathbf{1.18 \pm 0.33} \times 10^{-3}$ | $\mathbf{99.29 \pm 0.81}$ | $\mathbf{7.05 \pm 0.21} \times 10^{-4}$ |

*Table 2.* Performance comparison of MSNN and SNN on MNAR data, $\lambda = 0.05$.

| Algorithm | Low (Proportion $= (1.30 \pm 0.06)\%$) | | Medium (Proportion $= (1.49 \pm 0.06)\%$) | | High (Proportion $= (2.50 \pm 0.10)\%$) | |
|---|---|---|---|---|---|---|
| | FR % ($\uparrow$) | MRE ($\downarrow$) | FR % ($\uparrow$) | MRE ($\downarrow$) | FR % ($\uparrow$) | MRE ($\downarrow$) |
| SNN | $0.19 \pm 0.07$ | $0.349 \pm 0.139$ | $0.38 \pm 0.12$ | $0.390 \pm 0.143$ | $4.17 \pm 0.84$ | $0.351 \pm 0.050$ |
| MSNN (ours) | $\mathbf{3.13 \pm 0.41}$ | $\mathbf{0.117 \pm 0.006}$ | $\mathbf{3.26 \pm 0.34}$ | $\mathbf{0.114 \pm 0.007}$ | $\mathbf{4.52 \pm 0.62}$ | $\mathbf{0.106 \pm 0.004}$ |

*Table 3.* Performance comparison of MSNN and SNN on MNAR data, $\lambda = 0.02$.

| Algorithm | Low (Proportion $= (3.54 \pm 0.12)\%$) | | Medium (Proportion $= (3.76 \pm 0.12)\%$) | | High (Proportion $= (4.59 \pm 0.11)\%$) | |
|---|---|---|---|---|---|---|
| | FR % ($\uparrow$) | MRE ($\downarrow$) | FR % ($\uparrow$) | MRE ($\downarrow$) | FR % ($\uparrow$) | MRE ($\downarrow$) |
| SNN | $9.57 \pm 0.85$ | $0.366 \pm 0.027$ | $11.70 \pm 1.62$ | $0.379 \pm 0.037$ | $22.66 \pm 2.24$ | $0.383 \pm 0.025$ |
| MSNN (ours) | $\mathbf{26.96 \pm 3.50}$ | $\mathbf{0.129 \pm 0.008}$ | $\mathbf{33.88 \pm 4.20}$ | $\mathbf{0.135 \pm 0.010}$ | $\mathbf{54.16 \pm 4.30}$ | $\mathbf{0.118 \pm 0.009}$ |

# 6. Experiments

This section demonstrates MSNN's performance on both synthetic and real-world data. Experimental details including hyperparameters and real-world experiment background are provided in Appendix B. Here we compare MSNN to SNN and refer readers to Appendix C.1 for other baselines.

## 6.1. Simulation Study

Our theoretical analysis characterizes the expected number of valid anchors. In practice, enumerating multiple anchors is computationally expensive for large matrices; we therefore focus on the practically relevant regime $K_{(\cdot)} = 1$, and evaluate the feasibility probability $\mathbb{P}(K_{(\cdot)} \geq 1)$ predicted by our theory along with the prediction error.

**Experimental Setup.** Under our multi-treatment framework, we randomly generate latent factors $\boldsymbol{U} \in \mathbb{R}^{m \times r}$ and $\boldsymbol{V}^{(d)} \in \mathbb{R}^{n \times r}$, here $\boldsymbol{U}$ is normalized while $\boldsymbol{V}^{(d)}$ is scaled by a factor $f(d)$. For the treatment assignment, we focus on two settings: (i) MCAR, each entry is revealed at treatment level $d$ with $\mathbb{P}(D_{ij} = d) = p_{\mathrm{MCAR}}(d)$; (ii) MNAR, each entry at $(i, j)$ is revealed with $\mathbb{P}(D_{ij} = d) = p_{\mathrm{MNAR}}\left(A_{ij}^{d' \in \mathcal{L}}, d\right)$. For (ii) we take the multi-variant soft-max function $p_{\mathrm{MNAR}}\left(A_{ij}^{d' \in \mathcal{L}}, d\right) = \frac{e^{\lambda A_{ij}^d}}{\sum_{d' \neq 0} e^{\lambda A_{ij}^{d'}}}$ ($\lambda$ is a chosen hyper-parameter), simplifying the MNAR setting in (3) by fixing $\delta_d \equiv 0$, $\lambda_d \equiv \lambda$ and $s_0 = 0$.

Since the data that MSNN can use strictly covers all data that SNN can leverage and thus MSNN outperforms SNN, we mainly focus on cases that SNN fails, that is estimating data-scarce treatment levels. Following the imple-

mentation of Agarwal et al. (2023b), we filter out the invalid estimations by testing whether the (Mixed) Anchor Rows/Columns are feasible: approximate satisfaction of $x^{(k)}(d) \in \mathrm{Col}(S^{(k)}(d))$ (corresponding to Assumption 5.4) and $q^{(k)}(d) \in \mathrm{Row}(S^{(k)}(d))$ (corresponding to Assumption 3.4). We also filter out the estimations which (Mixed) Anchor Matrix is of size $(1, 1)$. We report both the feasible rate FR $:= \frac{\#\text{feasible estimated entries}}{mn}$ and mean relative estimation error MRE $:= \frac{\sum_{\hat{A}_{ij}^{(d)} \text{ is feasible}} \left| (\hat{A}_{ij}^{(d)} - A_{ij}^{(d)}) / f(d) \right|}{\#\text{feasible estimated entries}}$ conditioned on feasibility.

**Results.** We calculate the feasible ratio (FR) and mean relative error (MRE) of SNN and MSNN for all entries under data-scarce treatment levels ($d = $ low, medium, high). Results for MCAR and MNAR settings are reported in Tables 1, 2, and 3.

Across all settings, MSNN consistently achieves substantially higher feasible ratios than SNN, often nearly doubling the proportion of entries for which at least one valid anchor is found. Correspondingly, MSNN achieves lower estimation error, with MRE reduced by a factor of two to three for all treatment levels. This improvement arises from two complementary effects: MSNN increases the probability that at least one valid anchor exists, and, conditioned on feasibility, leverages substantially larger mixed anchor row and column sets than SNN. As a result, MSNN uses a larger effective sample size for reconstruction and attains lower estimation error.

These observations align with the theoretical sample efficiency analysis in Section 5.3. While the theory characterizes the expected number of anchors, it predicts a higher probability that $K_{(\cdot)} \geq 1$ under MSNN, which is exactly

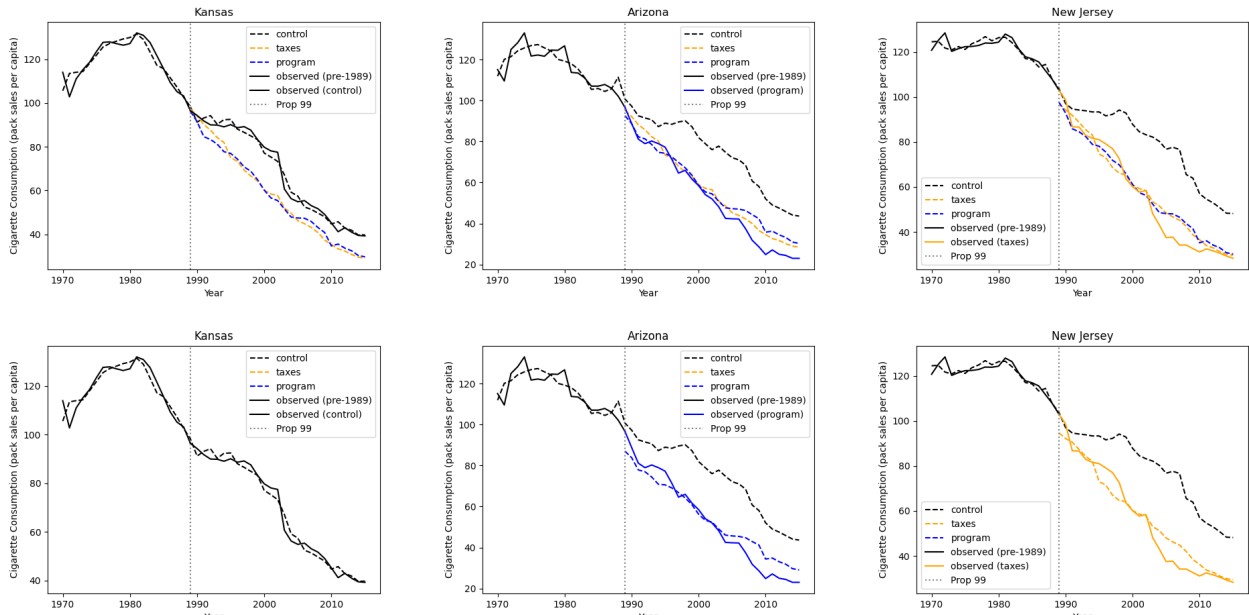

*Figure 2.* Selected prediction results of MSNN on Proposition 99 study in Abadie et al. (2010) compared to SNN. The three states Kansas, Arizona, New Jersey belong to treatment group of control, program, taxes respectively. The dashed lines are estimation results, while the solid lines indicates real-world observation. Before the year of 1989 (illustrated by the vertical dotted gray lines) all states are in control group so the solid lines are black-colored, after which their color varies. The dotted line indicates the time of Proposition 99 assignment. The dashed lines and solid lines of same color at the same time periods are close to each other, providing validation of the fitted outcomes.

reflected in the improved feasible ratio observed in practice. When the observed proportion of a treatment level falls below $2.5\%$, the data are too sparse for reliable estimation, and even MSNN can only recover a small fraction of entries. For treatment levels with observed proportions above $4\%$, MSNN operates efficiently with high feasibility and low error, whereas SNN continues to suffer from low feasible ratios and large estimation errors. This highlights that MSNN substantially enlarges the range of sparsity levels under which reliable estimation is possible.

**Robustness analysis.** Appendix C.2 varies the noise scale $\sigma$ from $1e-4$ to $0.1$. Appendix C.3 violates Assumption 3.5 by perturbing latent row factors $\boldsymbol{U}^{(d)}$ with scale $\zeta\boldsymbol{U}'^{(d)}$, demonstrating the robustness from an empirical perspective.

### 6.2. Case study: California Proposition 99

To exhibit the effect of our method on real-world model, we revisit the classic Proposition 99 study in Abadie et al. (2010). Different from Agarwal et al. (2020) who focuses on time-average treatment effect, we further estimate the counterfactual outcome at *each specific year* for each state.

In Figure 2 we select three states covering all treatment groups, comparing the estimation results of MSNN to SNN. In each sub-figure, the solid line indicates the observed outcome and dash lines indicate the estimated outcomes. The dash lines which have the same color with solid line at some

years indicates validation (for example, the black dash lines before 1989), while others indicate estimated counterfactual outcomes (not observed in reality). As illustrated by Figure 2, MSNN estimates more counterfactual outcomes than SNN. We also note that the results of our algorithm are consistent with the predicted counterfactual trends until the year of 2000 in the Figure 4 of Agarwal et al. (2020), though the latter does not discuss guaranteed error bound for each single year in their theoretical framework.

## 7. Conclusions and Future work

In this work we introduce a new entry-wise causal identification estimator named Mixed Synthetic Nearest Neighbors (MSNN), enabling cross-treatment data integration under the assumption that certain low-rank latent factors are shared across treatments. Theoretically, we prove that MSNN achieves an exponential improvement in sample efficiency for sparse treatment levels under both MCAR and MNAR settings, while preserving the finite-sample error bound and asymptotic normality guarantees of the original SNN estimator. Experiments on both synthetic and real-world data further demonstrate the efficacy of MSNN under limited data. We hope our work inspires future research in causal matrix completion under multiple treatment levels.

## Acknowledgment

Zhiheng Zhang is supported by "the Fundamental Research Funds for the Central Universities" (Grant No. 2025110602) of Shanghai University of Finance and Economics, and Independent Research Project (Grant No. 2026110081) funded by the School of Statistics and Data Science. This work was supported by the Shanghai Engineering Research Center of Finance Intelligence (Grant No. 19DZ2254600).

## Impact Statement

This paper presents work whose goal is to advance the field of Machine Learning. There are many potential societal consequences of our work, none which we feel must be specifically highlighted here.

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

# A. Related Work and Additional Discussion

## A.1. Related Work

This work is derived from our previous exploration (Zhang et al., 2020; Zhang & Su, 2024; Zhang, 2022; Zhang et al., 2023; Zhang, 2024; Su & Zhang et al, 2025; Wang et al., 2025; Zhang et al., 2025b; Zhang & Wang, 2025; Li et al., 2026; Su et al., 2023; 2024; 2026; Lu et al., 2026; Zhang, 2026) and we separate the literature review into the following parts.

**Matrix and Tensor Completion.** Classical matrix completion assumes data are missing completely at random (MCAR) or at random (MAR). (Candes & Recht, 2012; Keshavan et al., 2010) However in many real-world applications, missingness depends on the unobserved entries themselves, i.e., missing not at random (MNAR). (Schnabel et al., 2016; Ma & Chen, 2019) Existing work addresses this by jointly modeling the missingness mechanism and low-rank structure. (Ma & Chen, 2019; Sportisse et al., 2020; Yang et al., 2021) In the causal inference context, matrix completion can be used to estimate potential outcomes, as in Athey et al. (2021); Agarwal et al. (2023b;a).

Our work studies a general MNAR matrix completion problem under multiple treatments, which is formalized as a three-dimensional tensor completion framework aiming to achieve entry-wise treatment effect identification. Agarwal et al. (2023b) addresses the MNAR matrix completion problem under a binary treatment setup, and we show that our result strictly reduces to theirs in the binary case.

Building upon SNN (Agarwal et al., 2023b), Choi & Yuan (2024) introduces the CY estimator for nearest-neighbor-based matrix completion under multiple treatments and weak signal-to-noise ratios, while Sadhukhan et al. (2025) proposes two-sided nearest neighbors (TS-NN) that work without anchor rows/columns and beyond low-rank assumptions. Choi et al. (2025) proposes KernelNN to estimate counterfactual distributions rather than entry-wise values, and Dwivedi et al. (2024) develops a doubly robust nearest neighbors method for binary treatments.

Other approaches in causal matrix completion under multiple treatment levels have different frameworks from ours. Agarwal et al. (2020) adopts a similar three-dimensional tensor completion framework as ours, but one of the three dimensions is interpreted as time index $t$. Consequently, their goal is to estimate the time-average expected potential outcome during the post-treatment period, rather than entry-wise treatment effects. In contrast, our framework removes this interpretation and focuses on more general matrix completion setting, aiming to construct fine-grained, entry-wise treatment identification under multiple-treatment settings. Gao et al. (2025) also relies on time sequential data assumption different from our setting. They also utilize a gradient descent method different from ours. Zhang et al. (2025a) considers a generalized three-dimensional causal tensor completion framework under MNAR, where each entry is missing independently, which is different from our assumption that for an element at most one treatment level's potential outcome can be observed (the observations along the treatment dimension are mutually exclusive).

**Connection to Counterfactual Regression (CFR).** Classical CFR methods (Johansson et al., 2016; Shalit et al., 2017), aim to learn balanced representations so that treated and control populations become comparable. These approaches typically assume i.i.d. samples and focus on estimating individual treatment effects by reducing distributional imbalance between treatment groups.

In contrast, our setting involves structured missingness over matrices/tensors, where the observational pattern itself is non-random and highly dependent on latent factors. Rather than learning a balanced representation over i.i.d. units, our method addresses selection bias in a structured completion problem, where dependencies across rows/columns/treatments play a central role. Therefore, our work can be viewed as extending the spirit of CFR to a non-i.i.d., matrix/tensor completion regime, where balancing must be achieved jointly with low-rank structure learning.

**Connection to Causal Recommendation.** In recommendation systems, the observed data is often Missing Not At Random (MNAR) due to user self-selection and exposure bias. Prior works (Bonner & Vasile, 2018; Schnabel et al., 2016; Wang et al., 2021; Sato et al., 2020) explicitly formulate recommendation as a causal inference problem, where exposure is treated as the treatment and the goal is to debias logged interactions. Our framework is closely related in that we also interpret the observation mechanism as a treatment assignment process. However, unlike standard recommendation settings that operate on user–item interaction data, we consider a more general matrix/tensor completion problem with structured missingness, and our method leverages this structure to perform joint debiasing and completion.

Our framework treats these biases as a causal inference problem where "exposure" is the treatment. MSNN's use of Mixed Anchor Blocks serves as a debiasing mechanism, similar to how causal recommendation uses domain adaptation to align the distribution of biased logged data with the target population.

### A.2. Discussion, Justification and Robustness of Assumption 3.5

Assumption 3.5 is necessary for enabling cross-treatment data integration in our MSNN. It posits that each unit (row) possesses intrinsic, stable latent characteristics that remain invariant regardless of the specific treatment level applied, while treatment-specific loading matrices capture how the intervention interacts with these stable traits to produce outcomes.

**Standardness.** This assumption mirrors Assumption 2 of Agarwal et al. (2020) under a row–column exchange and does not impose stronger structural constraints. It is standard in low-rank causal models with structured treatments, also covering the low-rank tensor factorization model $Y_{ij}^{(d)} = \sum_{l \in [r]} u_{il} v_{jl} \lambda_{dl} + \epsilon_{ij}^{(d)}$ (mentioned in Agarwal et al. (2020)) as a special case by setting $v_{jl}^{(d)} = v_{jl} \lambda_{dl}$. Additionally, viewed as a tensor over $(i, j, d)$, Assumption 3.5 imposes a shared row-mode latent subspace across treatment slices, while allowing treatment-specific column loadings. This tensor-style shared-subspace structure is exactly what enables the cross-treatment transfer of imputation coefficients used by MSNN.

**Necessity.** This shared structure is the "bridge" that allows MSNN to aggregate information across treatments. If latent row factors were treatment-dependent (i.e., $U^{(d)}$ varies with $d$), observations from one treatment level would provide zero information about another, and the MSNN construction would collapse to the standard SNN framework, which cannot handle intrinsically sparse multi-treatment settings.

**Interpretability with examples.** For instance, in e-commerce pricing (rows = users, treatments = discount levels), a user's underlying preference and price sensitivity are stable traits ($U$). Different discounts affect purchase probabilities via treatment-specific factors ($V^{(d)}$), but both act upon the same latent user profile. In the Proposition 99 case study in section 6.2 (rows = states, columns = years), this assumption means intrinsic state-level factors affecting cigarette consumption remain stable, enabling consumption behavior from one treatment to inform another.

**Robustness analysis against moderate violations of Assumption 3.5.** Below, we also provide a theoretical explanation, showing the robustness of MSNN under moderate violation of Assumption 3.5.

Specifically, we show that the estimation error is linear to the perturbation scale under the following setting:

Consider that the latent row factors $U^{(d)}$ are perturbed by noise level $\zeta \in \mathbb{R}_+ \cup \{0\}$, i.e. $U^{(d)} = U_{\text{share}} + \zeta U_{\text{perturb}}^{(d)}$, where $U_{\text{share}}$ is shared and $U_{\text{perturb}}^{(d)}$ is noise, $U_{\text{perturb}}^{(d:d\in\mathcal{L})}$ and $U_{\text{share}}$ are i.i.d. Here, larger $\zeta$ indicates stronger perturbation.
*Remark* A.1. Theoretical justification of the linearity between the perturbation scale $\zeta$ and the estimation error.

We provide the following theoretical explanation by omitting the error matrix $E^{(d)}$. For fixed $U^{(d)}$, $U^{(d')}$s are linear to $\zeta$. Now $x$ is also fixed, from Algorithm 2, $A_{ij}^{(d)} = q(\zeta = 0)^{\top} S(\zeta = 0)^+ x$. Since $q$ and $S$ are constructed by directly concatenating submatrices of $A^{(d')} = U^{(d')}(\zeta) V^{(d')\top}$ for $d' \in \mathcal{L}$, $q$ and $S$ are linear to $\zeta$. Suppose $S(\zeta = 0)$ is not ill-conditioned, by Taylor's expansion, the deviation $q^{\top}(\zeta) S(\zeta)^+ x - A_{ij}^{(d)}$ is also **linear to $\zeta$ to the first order**.

As a supplement, Appendix C.3 also verifies this remark empirically.

### A.3. Discussion on the Conditioning on $\mathcal{E}$ in Asymptotic Normality in Theorem 5.6

*Remark* A.2 (Interpretation of conditioning on $\mathcal{E}$). The asymptotic normality in Theorem 5.6 is stated conditionally on the realized design event $\mathcal{E} = \{U, V, D\}$. More precisely, it should be understood along any sequence of problem instances indexed by the matrix size and the number of mixed anchor groups, for which the deterministic realization $\mathcal{E}$ satisfies the assumptions of the theorem. Formally, for some sequence $k \in \mathbb{N}_+$ and finite size event $\mathcal{E}_k$ satisfying the assumptions in Theorem 5.6, the error sequence satisfies the asymptotic normality. For simplicity, we omit the index $k$ here, following the expressions in Theorem 3 of the original SNN paper (Agarwal et al., 2023b).

# B. Experimental Details

This section further provides experimental details for Section 6.

**Hyperparameters of the simulation study.** In Section 6.1, the shape of the matrix is $m = 300$, $n = 100$, rank $r = 3$, the relative noise scale $\sigma$ is set to 0.001. Due to the small scale of experiments, the number of samples $K_{(.)}$ is set to 1. The treatment levels are $\mathcal{L} = \{\text{low, medium, high, very high}\}$, while the scales of treatment levels are $f(d) =$

$$
\begin{cases}
1 & , d = \text{low} \\
5 & , d = \text{medium} \\
25 & , d = \text{high} \\
625 & , d = \text{very high}
\end{cases}
.
$$

For MCAR simulation (i), the ground truth entries are within $[-f(d), f(d)]$, the probability distribution for each entry is

$$
p_{\text{MCAR}}(d) =
\begin{cases}
0.115 & , d = 0 \text{ (not observed)} \\
0.01 & , d = \text{low} \\
0.025 & , d = \text{medium} \\
0.05 & , d = \text{high} \\
0.8 & , d = \text{very high}
\end{cases}
$$

. For MNAR simulation (ii), we take the absolute value of each ground truth entry so that they are constrained within $[0, f(d)]$, the hyperparameter $\lambda$ in the probability distribution $p_{\text{MNAR}}(d)$ is set to 0.05 and 0.02.

For the feasible rate (FR) and mean relative estimation error (MRE), we report the mean and standard deviation on 10 repetitions.

**Background information of California Proposition 99 Data.** In 1988, California passed the Proposition 99 as the first large-scale anti-tobacco program, followed by other states which introduced similar programs (Arizona, Massachusetts, Oregon, and Florida) or raised tobacco taxes at least 50 cents (Alaska, Hawaii, Maryland, Michigan, New Jersey, New York, Washington). The other 38 states remain status quo, which are treated as the control group. One natural question is that what if a state remains status quo had raised tobacco taxes or introduced anti-tobacco program, or what if a state introduced anti-tobacco program had remained status quo or raised tobacco taxes etc., that is estimating the counterfactual outcomes, as discussed in Agarwal et al. (2023a; 2020).

Following the experimental settings in Agarwal et al. (2020), we leverage the data from both pre-treatment time and post-treatment time. For simplicity we approximately treat the years after 1989 as post-treatment time. Here we manually set $K = 1$, and $w(d) \equiv const$ for all treatment levels.

*Table 4.* Performance comparison of SNN, CY, TS-NN and MSNN on MCAR data.

| Algorithm | **Low** ($p(d) = 0.01$) | | **Medium** ($p(d) = 0.025$) | | **High** ($p(d) = 0.05$) | |
|---|---|---|---|---|---|---|
| | FR % ($\uparrow$) | MRE ($\downarrow$) | FR % ($\uparrow$) | MRE ($\downarrow$) | FR % ($\uparrow$) | MRE ($\downarrow$) |
| SNN | $0.02 \pm 0.02$ | $0.806 \pm 0.240$ | $1.20 \pm 0.36$ | $0.577 \pm 0.110$ | $11.34 \pm 0.82$ | $0.515 \pm 0.046$ |
| CY | $100.00 \pm 0.00$ | $0.499 \pm 0.002$ | $100.00 \pm 0.00$ | $0.494 \pm 0.001$ | $100.00 \pm 0.00$ | $0.481 \pm 0.002$ |
| TS-NN | $0.92 \pm 0.16$ | $0.353 \pm 0.024$ | $19.83 \pm 2.66$ | $0.508 \pm 0.014$ | $92.19 \pm 2.04$ | $0.467 \pm 0.009$ |
| MSNN (ours) | $4.69 \pm 1.11$ | $(\mathbf{3.91 \pm 1.09}) \times 10^{-2}$ | $63.73 \pm 5.67$ | $(\mathbf{1.18 \pm 0.33}) \times 10^{-3}$ | $99.29 \pm 0.81$ | $(\mathbf{7.05 \pm 0.21}) \times 10^{-4}$ |

*Table 5.* Performance comparison of SNN, CY, TS-NN and MSNN on MNAR data, $\lambda = 0.05$.

| Algorithm | **Low** (Proportion $= (0.02 \pm 0.01)\%$) | | **Medium** (Proportion $= (0.02 \pm 0.01)\%$) | | **High** (Proportion $= (0.04 \pm 0.02)\%$) | |
|---|---|---|---|---|---|---|
| | FR % ($\uparrow$) | MRE ($\downarrow$) | FR % ($\uparrow$) | MRE ($\downarrow$) | FR % ($\uparrow$) | MRE ($\downarrow$) |
| SNN | $0.00 \pm 0.00$ | $-$ | $0.00 \pm 0.00$ | $-$ | $0.00 \pm 0.00$ | $-$ |
| CY | $100.00 \pm 0.00$ | $0.749 \pm 0.009$ | $100.00 \pm 0.00$ | $0.749 \pm 0.006$ | $100.00 \pm 0.00$ | $0.751 \pm 0.006$ |
| TS-NN | $60.01 \pm 51.63$ | $(1.288 \pm 1.109) \times 10^9$ | $70.01 \pm 48.29$ | $(3.006 \pm 2.075) \times 10^8$ | $10.02 \pm 31.61$ | $(0.8590 \pm 2.716) \times 10^7$ |
| MSNN (ours) | $1.60 \pm 0.84$ | $\mathbf{0.148 \pm 0.038}$ | $1.24 \pm 0.77$ | $\mathbf{0.151 \pm 0.046}$ | $2.52 \pm 0.93$ | $\mathbf{0.149 \pm 0.040}$ |

*Table 6.* Performance comparison of SNN, CY, TS-NN and MSNN on MNAR data, $\lambda = 0.02$.

| Algorithm | **Low** (Proportion $= (0.29 \pm 0.05)\%$) | | **Medium** (Proportion $= (0.29 \pm 0.06)\%$) | | **High** (Proportion $= (0.37 \pm 0.09)\%$) | |
|---|---|---|---|---|---|---|
| | FR % ($\uparrow$) | MRE ($\downarrow$) | FR % ($\uparrow$) | MRE ($\downarrow$) | FR % ($\uparrow$) | MRE ($\downarrow$) |
| SNN | $0.04 \pm 0.04$ | $0.118 \pm 0.055$ | $0.05 \pm 0.04$ | $0.170 \pm 0.096$ | $0.11 \pm 0.07$ | $0.244 \pm 0.273$ |
| CY | $100.00 \pm 0.00$ | $0.748 \pm 0.009$ | $100.00 \pm 0.00$ | $0.748 \pm 0.006$ | $100.00 \pm 0.00$ | $0.749 \pm 0.006$ |
| TS-NN | $0.23 \pm 0.13$ | $\mathbf{0.080 \pm 0.039}$ | $0.25 \pm 0.09$ | $\mathbf{0.072 \pm 0.030}$ | $0.42 \pm 0.23$ | $\mathbf{0.075 \pm 0.022}$ |
| MSNN (ours) | $16.33 \pm 1.92$ | $0.142 \pm 0.014$ | $16.57 \pm 1.60$ | $0.143 \pm 0.010$ | $18.63 \pm 1.97$ | $0.136 \pm 0.014$ |

*Table 7.* Performance comparison of SNN, CY, TS-NN and MSNN on MNAR data, $\lambda = 0.01$.

| Algorithm | **Low** (Proportion $= (1.87 \pm 0.15)\%$) | | **Medium** (Proportion $= (1.97 \pm 0.20)\%$) | | **High** (Proportion $= (2.23 \pm 0.23)\%$) | |
|---|---|---|---|---|---|---|
| | FR % ($\uparrow$) | MRE ($\downarrow$) | FR % ($\uparrow$) | MRE ($\downarrow$) | FR % ($\uparrow$) | MRE ($\downarrow$) |
| SNN | $4.32 \pm 1.52$ | $0.228 \pm 0.022$ | $4.93 \pm 1.50$ | $0.219 \pm 0.025$ | $7.93 \pm 2.59$ | $0.232 \pm 0.038$ |
| CY | $100.00 \pm 0.00$ | $0.739 \pm 0.009$ | $100.00 \pm 0.00$ | $0.739 \pm 0.006$ | $100.00 \pm 0.00$ | $0.738 \pm 0.006$ |
| TS-NN | $12.25 \pm 4.43$ | $0.135 \pm 0.013$ | $14.09 \pm 4.79$ | $0.138 \pm 0.012$ | $20.41 \pm 6.20$ | $0.142 \pm 0.010$ |
| MSNN (ours) | $68.82 \pm 2.11$ | $\mathbf{0.086 \pm 0.010}$ | $69.84 \pm 2.39$ | $\mathbf{0.083 \pm 0.016}$ | $73.73 \pm 2.79$ | $\mathbf{0.074 \pm 0.013}$ |

## C. Additional Experiments

### C.1. Result on Other Baselines

This section enriches the results in Section 6.1 by adding baselines CY (Choi & Yuan, 2024) and TS-NN (Sadhukhan et al., 2025). Table 4 represents the result under MCAR data, while Table 5, 6 and 7 represent the result under MNAR data. The settings of Table 4 follows the experimental settings in Section B, which is consistent with Table 1, while Table 5, 6 and 7 change the data generation procedure in Section B: the range of $U$ and $V(d)$ are adjusted to non-negative values, so that the underlying ground truth entries are within $[0, f(d)]$.

From the results we observe that both CY and TS-NN improve the SNN, but still fail under the multiple treatment data, either with low feasible rate or large estimation error. This is mostly because they are not particularly designed for estimation in data-scarce treatment level under multi-treatment.

In particular, CY applies nuclear-norm regularized matrix completion at the matrix or block level, rather than using an entry-wise anchor-based feasibility criterion as in SNN. Therefore, CY produces an estimate for every target entry in our experiments, and its feasible rate is reported as $100\%$. However, this does not imply that all target entries are equally well identified; rather, the estimation error can still be large when a treatment level contains insufficient observed information.

### C.2. Sensitivity Analysis on the Noise Scale $\sigma$

In Table 8, we vary the noise scale from $1e-4$ to $1e-1$, and estimate the potential outcomes of medium treatment level (with probability $p(d = \text{medium}) = 0.025$) under MCAR. The feasible ratios remain stable for $\sigma$ under $1e-2$, while the

*Table 8.* Sensitivity analysis on relative noise scale $\sigma$ on MCAR data, MSNN.

| $\sigma$ | **Medium** $(p(d) = 0.025)$ | |
| --- | --- | --- |
| | FR % ($\uparrow$) | MRE ($\downarrow$) |
| 0.1 | $43.97 \pm 5.96$ | $(\mathbf{74.69 \pm 2.81}) \times 10^{-3}$ |
| 0.01 | $63.72 \pm 5.67$ | $(\mathbf{7.99 \pm 0.54}) \times 10^{-3}$ |
| 0.001 | $63.73 \pm 5.67$ | $(\mathbf{1.18 \pm 0.33}) \times 10^{-3}$ |
| 0.0001 | $63.73 \pm 5.67$ | $(\mathbf{0.50 \pm 0.32}) \times 10^{-3}$ |

*Table 9.* Robustness analysis of MSNN under the violation of Assumption 3.5 on MCAR data.

| $\zeta$ | **Medium** $(p(d) = 0.025)$ | |
| --- | --- | --- |
| | FR % ($\uparrow$) | MRE ($\downarrow$) |
| 0.0 | $63.73 \pm 5.67$ | $(1.18 \pm 0.33) \times 10^{-3}$ |
| 0.0001 | $63.31 \pm 5.84$ | $(1.31 \pm 0.29) \times 10^{-3}$ |
| 0.001 | $63.31 \pm 5.84$ | $(1.79 \pm 0.33) \times 10^{-3}$ |
| 0.01 | $63.22 \pm 5.83$ | $(9.84 \pm 1.04) \times 10^{-3}$ |
| 0.02 | $63.22 \pm 5.83$ | $(1.90 \pm 0.17) \times 10^{-2}$ |
| 0.04 | $63.22 \pm 5.83$ | $(3.71 \pm 0.29) \times 10^{-2}$ |
| 0.06 | $62.83 \pm 5.85$ | $(5.49 \pm 0.39) \times 10^{-2}$ |
| 0.08 | $62.22 \pm 6.00$ | $(7.22 \pm 0.47) \times 10^{-2}$ |
| 0.10 | $61.17 \pm 6.09$ | $(8.88 \pm 0.47) \times 10^{-2}$ |

mean relative estimation errors grow linearly with $\sigma$. This shows that the estimation error does not explode and steadily grows as the noise scale increases, giving that our estimation is robust against noise. Other hyperparameters follow the experimental settings in Section B.

## C.3. Robustness Analysis of MSNN under the Violation of Assumption 3.5

This section provides empirical evidence on the robustness of MSNN against moderate violations of Assumption 3.5.

Following the settings in Remark A.1, we perturb the latent row factors $\boldsymbol{U}^{(d)}$ by $\boldsymbol{U}^{(d)} = \boldsymbol{U}_{\text{share}} + \zeta \boldsymbol{U}^{(d)}_{\text{perturb}}$, where $\boldsymbol{U}_{\text{share}}$ is shared and $\zeta \boldsymbol{U}^{(d)}_{\text{perturb}}$ is noise. We vary $\zeta$ from $1e-4$ to $1e-1$, and compare the feasible ratios and mean relative estimation errors on MCAR data in Table 9. Other experimental details follow the settings in Section B.

The result in Table 9 shows that the estimation error grows linearly with the noise level $\zeta$, supporting the theoretical analysis in Remark A.1 and showing robustness under moderate violation.

# D. Proofs of Theorems and Supporting Lemmas

## D.1. Proof of Identification

Proof of Lemma 3.6:

*Proof.* The proof is straightforward:

$$
\begin{aligned}
u_i^{(d')} &\overset{A3.5}{=} u_i^{(d)} && \because \text{Assumption 3.5} \\
&\overset{A3.4}{=} \sum_{l \in \mathcal{I}^{(d)}(i)} \beta_l^{(d)}\left(\mathcal{I}^{(d)}(i)\right) u_l^{(d)} && \because \text{Assumption 3.4} \\
&\overset{A3.5}{=} \sum_{l \in \mathcal{I}^{(d)}(i)} \beta_l^{(d)}\left(\mathcal{I}^{(d)}(i)\right) u_l^{(d')}. && \because \text{Assumption 3.5}
\end{aligned}
\tag{4}
$$

□

Proof of Theorem 3.7:

*Proof.*

$$
\begin{aligned}
A_{ij}^{(d)} &= \mathbb{E}\left[Y_{ij}^{(d)}\middle| \boldsymbol{U}, \boldsymbol{V}\right] \\
&\overset{A3.1}{=} \mathbb{E}\left[\left\langle u_i^{(d)}, v_j^{(d)}\right\rangle + \epsilon_{ij}^{(d)}\middle| \boldsymbol{U}, \boldsymbol{V}\right] && \because \text{Assumption 3.1} \\
&\overset{A3.2}{=} \left\langle u_i^{(d)}, v_j^{(d)}\right\rangle\middle| \boldsymbol{U}, \boldsymbol{V} && \because \text{Assumption 3.2} \\
&= \left\langle u_i^{(d)}, v_j^{(d)}\right\rangle\middle| \boldsymbol{U}, \boldsymbol{V}, \boldsymbol{D} \\
&\overset{L3.6}{=} \sum_{l \in \mathcal{I}(i)} \beta_l\left(\mathcal{I}(i)\right)\left\langle u_l^{(d)}, v_j^{(d)}\right\rangle\middle| \boldsymbol{U}, \boldsymbol{V}, \boldsymbol{D} && \because \text{Lemma 3.6} \\
&\overset{A3.2}{=} \sum_{l \in \mathcal{I}(i)} \beta_l\left(\mathcal{I}(i)\right)\mathbb{E}\left[\left\langle u_l^{(d)}, v_j^{(d)}\right\rangle + \epsilon_{lj}^{(d)}\middle| \boldsymbol{U}, \boldsymbol{V}, \boldsymbol{D}\right] && \because \text{Assumption 3.2} \\
&\overset{A3.1}{=} \sum_{l \in \mathcal{I}(i)} \beta_l\left(\mathcal{I}(i)\right)\mathbb{E}\left[Y_{lj}^{(d)}\middle| \boldsymbol{U}, \boldsymbol{V}, \boldsymbol{D}\right] && \because \text{Assumption 3.1} \\
&= \sum_{l \in \mathcal{I}(i)} \beta_l\left(\mathcal{I}(i)\right)\mathbb{E}\left[\tilde{Y}_{lj}\middle| \boldsymbol{U}, \boldsymbol{V}, \boldsymbol{D}\right].
\end{aligned}
\tag{5}
$$

The last step is from $D_{lj,l \in \mathcal{I}(i)} = d$, which gives $\tilde{Y}_{lj} = Y_{lj}^{(d)}, \forall l \in \mathcal{I}(i)$.

□

## D.2. Proof of Finite-sample Bound and Asymptotic Normality

Before proceeding on the proofs of Theorem 5.5 and 5.6, we provide a simple illustration under zero-noise condition:

**Theorem D.1.** *For Algorithm 2, under Assumptions 3.1, 3.5, 3.2, 3.4, 5.4, given $(i, j, d)$ with subgroup $(k)$. If $\left|\mathrm{MAR}^{(k)}(d)\right| \geq \mu$, then*

$$
A_{ij}^{(d)} = \mathbb{E}\left[q_w^{(k)}(d)\middle|\mathcal{E}\right] \mathbb{E}\left[\boldsymbol{S}_w^{(k)}(d)\middle|\mathcal{E}\right]^+ \cdot \mathbb{E}\left[x^{(k)}(d)\middle|\mathcal{E}\right],
\tag{6}
$$

*where $\cdot^+$ denotes pseudo-inverse.*

*To rephrase, let $\lambda^{(k)}(d) = \mathrm{rank}\left(\mathbb{E}\left[\boldsymbol{S}_w^{(k)}(d)\middle|\mathcal{E}\right]\right)$, if $\epsilon_{i'j'}^{(d')} = 0$ a.s. for all $(i', j', d') \in [m] \times [n] \times \mathcal{L}$, then $\hat{A}_{ij,k}^{(d)} = A_{ij}^{(d)}$.*

*Proof.* Under Theorem 3.7, there exists $\beta \in \mathbb{R}^{\mathrm{MAR}^{(k)}(d)}$ s.t. for all $b \in \mathrm{MAC}^{(k)}(d) \cup \{j\}$, $A_{ib}^{(d(b))} = \sum_{l \in \mathrm{MAR}^{(k)}(d)} \beta_l \mathbb{E}\left[\tilde{Y}_{lb}\middle|\mathcal{E}\right]$.

Then $\mathbb{E}\left[q_w^{(k)}(d)\middle|\mathcal{E}\right] = \beta^\top \mathbb{E}\left[\boldsymbol{S}_w^{(k)}(d)\middle|\mathcal{E}\right]$, $A_{ij}^{(d)} = \beta^\top \mathbb{E}\left[x^{(k)}(d)|\mathcal{E}\right]$.

From Assumption 5.4, there exists some $\eta \in \mathbb{R}^{\mathrm{MAC}^{(k)}(d)}$ s.t.

$$\mathbb{E}\left[x^{(k)}(d)\middle|\mathcal{E}\right] = \mathbb{E}\left[\boldsymbol{S}_w^{(k)}(d)\middle|\mathcal{E}\right]\eta. \tag{7}$$

Then for any such $\eta$, we have

$$A_{ij}^{(d)} = \beta^\top \mathbb{E}\left[x^{(k)}(d)\middle|\mathcal{E}\right] = \beta^\top \mathbb{E}\left[\boldsymbol{S}_w^{(k)}(d)\middle|\mathcal{E}\right]\eta = \mathbb{E}\left[q_w^{(k)}(d)\middle|\mathcal{E}\right]\eta. \tag{8}$$

Let $\eta = \mathbb{E}\left[\boldsymbol{S}_w^{(k)}(d)\middle|\mathcal{E}\right]^+ \mathbb{E}\left[x^{(k)}(d)|\mathcal{E}\right]$, where $\cdot^+$ denotes pseudo-inverse, clearly $\mathbb{E}\left[\boldsymbol{S}_w^{(k)}(d)\middle|\mathcal{E}\right]\eta = \mathcal{P}_{\mathbb{E}\left[\boldsymbol{S}_w^{(k)}(d)\middle|\mathcal{E}\right]} \mathbb{E}\left[x^{(k)}(d)\middle|\mathcal{E}\right] = \mathbb{E}\left[x^{(k)}(d)\middle|\mathcal{E}\right]$, where $\mathcal{P}_{\mathbb{E}\left[\boldsymbol{S}_w^{(k)}(d)\middle|\mathcal{E}\right]}$ is the projection onto $\mathrm{Col}\left(\mathbb{E}\left[\boldsymbol{S}_w^{(k)}(d)\middle|\mathcal{E}\right]\right)$. Then the proof is completed.

$\square$

Now we prove Theorem 5.5 and 5.6:

*Proof.* Suppose we aim to estimate $A_{ij}^{(d)}$ a fixed unit $(i,j)$ and treatment level $d$, and we obtain $K_{\mathrm{MSNN}}$ subgroups of $\boldsymbol{S}_w^{(k)}(d)$, $q_w^{(k)}(d)$ and $x^{(k)}(d)$ ($k \in [K_{\mathrm{MSNN}}]$).

Given treatment assignment $\boldsymbol{D}$ with the scaling factor $f(d)$, we define the scaling matrix $\boldsymbol{F} := \left[\frac{1}{f(D_{ij})}\right]_{i \in [m], j \in [n]}$. (For $D_{ij} = 0$, we explicitly set $\boldsymbol{F}_{ij} = 1$. Such entries are not used afterwards so this does not affect the proof.) We further define the normalized latent factor as $\boldsymbol{V}^{(d)\prime} := \frac{1}{f(d)}\boldsymbol{V}^{(d)}$, and similarly the normalized error term $\boldsymbol{E}^{(d)\prime} = \frac{1}{f(d)}\boldsymbol{E}^{(d)}$. Since we assume that $\mathbb{E}\left[\left(\epsilon_{ij}^{(d)}\right)^2\right] = \left(\sigma_{ij}^{(d)}\right)^2$ and $\left\|\epsilon_{ij}^{(d)}\right\|_{\psi_2} \le C\sigma_{ij}^{(d)}$, where $\sigma_{ij}^{(d)} \le f(d)\sigma$, then for $\boldsymbol{E}'$ we have $\mathbb{E}\left[\left(\epsilon_{ij}^{(d)\prime}\right)^2\right] = \left(\sigma_{ij}^{(d)\prime}\right)^2$ and $\left\|\epsilon_{ij}^{(d)\prime}\right\|_{\psi_2} \le C\sigma_{ij}^{(d)\prime}$, where $\sigma_{ij}^{(d)\prime} \le \sigma$.

For observed outcome $\tilde{Y}$, we construct $\tilde{Y}' = \tilde{Y} \odot \boldsymbol{F}$, where $\odot$ denotes entry-wise product (Hadamard product). Given $\boldsymbol{S}_w^{(k)}(d)$, $q_w^{(k)}(d)$ and $x^{(k)}(d)$, we denote the submatrices/vectors at the same positions of $\tilde{Y}'$ as $\boldsymbol{S}^{(k)\prime}$, $q^{(k)\prime}$ and $x^{(k)\prime}$ respectively:

$$\begin{aligned}
\boldsymbol{S}^{(k)\prime} &= \left[\tilde{Y}'_{ab} : (a,b) \in \mathrm{MAR}^{(k)}(d) \times \mathrm{MAC}^{(k)}(d)\right], \\
q^{(k)\prime} &= \left[\tilde{Y}'_{ib}\right]_{b \in \mathrm{MAC}^{(k)}(d)}, \\
x^{(k)\prime} &= \left[\tilde{Y}'_{aj}\right]_{a \in \mathrm{MAR}^{(k)}(d)}.
\end{aligned} \tag{9}$$

Now from definition, $\boldsymbol{S}_w^{(k)}(d) = \boldsymbol{S}^{(k)\prime}$, $q_w^{(k)}(d) = q^{(k)\prime}$, $x^{(k)}(d) = f(d) \cdot x^{(k)\prime}$.

Thus if we apply SNN (Algorithm 1) to $\boldsymbol{S}^{(k)\prime}$, $q^{(k)\prime}$ and $x^{(k)\prime}$ to obtain another estimation $\hat{A}_{ij,k}^{(d)\prime}$ (though now we do not know its meaning), then the estimation $\hat{A}_{ij,k}^{(d)}$ satisfy:

$$\hat{A}_{ij,k}^{(d)} = f(d)\hat{A}_{ij,k}^{(d)\prime}. \tag{10}$$

Now from Theorem 2 and 3 of Agarwal et al. (2023b), it remains us to prove that $\boldsymbol{S}^{(k)\prime}$, $q^{(k)\prime}$ and $x^{(k)\prime}$ comes from another low-rank factor model. We construct the model as the following procedure:

- As the first step, we select $\boldsymbol{V}''$ to be $\boldsymbol{V}^{(d)\prime}$, where $d$ is the estimated treatment level mentioned at the beginning. Then for all $k \in [K_{\mathrm{MSNN}}]$, for all $b \in \mathrm{MAC}^{(k)}(d)$, replace the $b^{th}$ column of $\boldsymbol{V}''$ by the $b^{th}$ column of $\boldsymbol{V}^{(\bar{d}(b))\prime}$. (Notice that from the construction of subgroups $k \in [K_{\mathrm{MSNN}}]$, $d(b) := D_{ab:a \in \mathrm{MAR}^{(k)}(d), b \in \mathrm{MAC}^{(k)}(d)} = D_{ib:b \in \mathrm{MAC}^{(k)}(d)}$. )

- We conduct similar procedure for the error matrix: first we set $\boldsymbol{E}'' = \boldsymbol{E}^{(d)\prime}$, then for all $k \in [K_{\mathrm{MSNN}}]$ and for all $b \in \mathrm{MAC}^{(k)}(d)$, replace the $b^{th}$ column of $\boldsymbol{E}''$ by the $b^{th}$ column of $\boldsymbol{E}^{(d(b))\prime}$. (Now the entries of $\boldsymbol{E}''$ are still independent. )

- Finally, we set $\boldsymbol{A}'' = \boldsymbol{U}\boldsymbol{V}''^{\top}$, $\tilde{\boldsymbol{Y}}'' = \boldsymbol{A}'' + \boldsymbol{E}''$.

Now $\tilde{\boldsymbol{Y}}'' = \boldsymbol{U}\boldsymbol{V}''^{\top} + \boldsymbol{E}''$ satisfy the assumptions of low-rank factor model discussed in Agarwal et al. (2023b), while the entries of $\boldsymbol{A}''$ are bounded by $[-1, 1]$ conditioned on $\mathcal{E}$. From the construction of $\boldsymbol{S}_w^{(k)}(d)$, $q_w^{(k)}(d)$ and $x^{(k)}(d)$, $\tilde{\boldsymbol{Y}}''$ and $\tilde{\boldsymbol{Y}}'$ are exactly the same on the positions of $\boldsymbol{S}_w^{(k)}(d)$, $q_w^{(k)}(d)$ and $x^{(k)}(d)$. This reduces our proof to the Theorem 2 and 3 of Agarwal et al. (2023b) and thus completes the proof of finite-sample bound and asymptotic normality.

$\square$

Finally, we consider the case that $f(d)$ is not accurately obtained but only estimated by $\hat{f}(d)$, and see how this affects the finite sample bound and the asymptotic normality.

*Remark* D.2 (Estimated treatment-scale weights, detailed explanation). Theorems 5.5 and 5.6 are stated for the ideal choice $w(b, d(b)) = 1/f(d(b))$. However in some cases, the scales $f(d)$ are unknown. Suppose we use estimated positive scales $\hat{f}(d)$ instead, i.e., $w(b, d(b)) = 1/\hat{f}(d(b))$ in estimation, while $f(d)/\hat{f}(d)$ is upper&lower bounded by some constants for all treatments $\forall d \in \mathcal{L}$.

Consider the rescaled potential outcome $A_{ij}^{\star(d)} := A_{ij}^{(d)} / \max_{d' \in \mathcal{L}}(f(d')/\hat{f}(d'))$ and noise $\epsilon_{ij}^{\star(d)} := \epsilon_{ij}^{(d)} / \max_{d' \in \mathcal{L}}(f(d')/\hat{f}(d'))$, then Assumptions 5.1-5.4 hold for the rescaled model with estimated scale $\hat{f}(d)$. Therefore, the proof of Theorem 5.5 applies verbatim, with the leading scale factor $f(d)$ replaced by $\hat{f}(d) \cdot \max_{d' \in \mathcal{L}}(f(d')/\hat{f}(d'))$. The asymptotic normality has the same limiting normalization as in Theorem 5.6: the negligibility condition (iv) is modified only by a constant factor, so the same asymptotic normality remains valid.

### D.3. Proof of Data Efficiency

This section proves the Theorems and Corollaries in Section 5.3. At the beginning, we establish two key lemmas in Section D.3.1, then prove the results of MCAR/MNAR in Section D.3.2 and D.3.3, then prove the lemmas in Section D.3.4.

For simplicity, below we denote the $K'_{(\cdot)}(d)$ to be the number of subgroups with the constraint of disjoint $x^{(k)}(d)$ removed, $K^p_{(\cdot)}(d)$ to be the number of pairs of subgroups $k_1$, $k_2$ that $x^{(k_1)} \cap x^{(k_2)} \neq \emptyset$.

#### D.3.1. KEY SUPPORTING LEMMAS

We bound the expectation of the maximum number of samples $\mathbb{E}[K_{(\cdot)}]$ by the following lemma:

**Lemma D.3.** *Given any entry $(i, j)$ under treatment $d$, for $|\mathrm{AR}(d)| = |\mathrm{MAR}(d)| = r$, $|\mathrm{AC}(d)| = |\mathrm{MAC}(d)| = c$, the expectation of its $K_{\mathrm{SNN}}$ and $K_{\mathrm{MSNN}}$ can be upper/lower bounded by:*

$$\mathbb{E}\left[K_{(\cdot)}(d)\right] / \mathbb{E}\left[K'_{(\cdot)}(d)\right] \in \left[\left(1 + \frac{2\mathbb{E}\left[K^p_{(\cdot)}(d)\right]}{\mathbb{E}\left[K'_{(\cdot)}(d)\right]}\right)^{-1}, 1\right]. \tag{11}$$

We also provide the following Lemma on the summation of combinatorial numbers:

**Lemma D.4.** *Summation of combinatorial numbers.*

*For $p_1, p_2, p_3 > 0$, $p_3 \leq 1$, we have*

$$
\sum_{\substack{0 \leq r' \leq r-1, \\ 0 \leq c' \leq c, \\ (r', c') \neq (0,0)}} \binom{M}{r'} \binom{r}{r'} \binom{N}{c'} \binom{c}{c'} p_1^{r'} p_2^{(r+1)c'} p_3^{r'(c-c')}
$$

$$
\leq \left( I_0 \left( 2\sqrt{Mrp_1 p_3^c} \right) - 1 \right) + \left( I_0 \left( 2\sqrt{Ncp_2^{r+1}} \right) - 1 \right)
$$

$$
+ \left( I_0 \left( 2\sqrt{Mrp_1 p_3^{\alpha c}} \right) - 1 \right) \left( I_0 \left( 2\sqrt{Ncp_2^{r+1} p_3^{\alpha(1-r)}} \right) - 1 \right).
$$

(12)

*where $\alpha \in (0, 1)$ is an arbitrary constant, $I_0(x) := \sum_{k=0}^{+\infty} \frac{(x/2)^{2k}}{(k!)^2}$ is the Modified Bessel Function of the First Kind, Order Zero.*

The proofs of Lemma D.3 and D.4 are provided in Section D.3.4, after the proof of main theorems.

D.3.2. DATA EFFICIENCY UNDER MISSING COMPLETELY AT RANDOM

This section proves Theorem 5.8 and Corollary 5.10, 5.11 in section 5.3.1.

We first prove Theorem 5.8 under a relaxed sparsity condition:

$$
mrp_d p_{\max}^{\alpha c} < 1, \ nc\gamma p_{\max}^{(1-\alpha)r+1+\alpha} < 1.
$$

(13)

*Proof.* For simplicity, we omit $(d)$ in $K_{(\cdot)}(d)$, $K'_{(\cdot)}(d)$ and $K^p_{(\cdot)}(d)$ during the proof.

For the upper bound, we consider $\mathbb{E}\left[ K'_{(\cdot)} \right]$:

$$
\mathbb{E}\left[ K_{\mathrm{SNN}} \right] \leq \mathbb{E}\left[ K'_{\mathrm{SNN}} \right]
$$

$$
= \mathbb{E}\left[ \sum_{\substack{\mathrm{AR}(d) \subseteq [m] \setminus i, \\ \mathrm{AC}(d) \subseteq [n] \setminus j}} \mathbf{1}\{ D_{ab, a \in \mathrm{AR}(d), b \in \mathrm{AC}(d)} = D_{aj, a \in \mathrm{AR}(d)} = D_{ib, b \in \mathrm{AC}(d)} = d \} \right]
$$

$$
= \sum_{\substack{\mathrm{AR}(d) \subseteq [m] \setminus i, \\ \mathrm{AC}(d) \subseteq [n] \setminus j}} \mathbb{E}\left[ \mathbf{1}\{ D_{ab, a \in \mathrm{AR}(d), b \in \mathrm{AC}(d)} = D_{aj, a \in \mathrm{AR}(d)} = D_{ib, b \in \mathrm{AC}(d)} = d \} \right]
$$

$$
= \sum_{\substack{\mathrm{AR}(d) \subseteq [m] \setminus i, \\ \mathrm{AC}(d) \subseteq [n] \setminus j}} p_d^{rc+r+c} = \binom{m-1}{r} \binom{n-1}{c} p_d^{rc+r+c},
$$

(14)

and

$$\mathbb{E}\left[K_{\text{MSNN}}\right] \leq \mathbb{E}\left[K'_{\text{MSNN}}\right]$$

$$=\mathbb{E}\left[\sum_{\substack{\text{MAR}(d)\subseteq[m]\setminus i,\\ \text{MAC}(d)\subseteq[n]\setminus j}} \mathbf{1}\{D_{ab,a\in\text{MAR}(d),b\in\text{MAC}(d)} = D_{ib,b\in\text{MAC}(d)}, D_{aj,a\in\text{MAR}(d)} = d\}\right]$$

$$=\sum_{\substack{\text{MAR}(d)\subseteq[m]\setminus i,\\ \text{MAC}(d)\subseteq[n]\setminus j}} \mathbb{E}\left[\mathbf{1}\{D_{ab,a\in\text{MAR}(d),b\in\text{MAC}(d)} = D_{ib,b\in\text{MAC}(d)}, D_{aj,a\in\text{MAR}(d)} = d\}\right] \tag{15}$$

$$=\sum_{\substack{\text{MAR}(d)\subseteq[m]\setminus i,\\ \text{MAC}(d)\subseteq[n]\setminus j}} p_d^r \cdot \mathbb{P}\left(D_{ab,a\in\text{MAR}(d),b\in\text{MAC}(d)} = D_{ib,b\in\text{MAC}(d)}\right)$$

$$=\sum_{\substack{\text{MAR}(d)\subseteq[m]\setminus i,\\ \text{MAC}(d)\subseteq[n]\setminus j}} \left[p_d^r \cdot \left(\sum_{d'\in\mathcal{L}} p_{d'}^{r+1}\right)^c\right] = \binom{m-1}{r}\binom{n-1}{c}\gamma^c p_d^r p_{\max}^{(r+1)c}.$$

Then we upper bound the number of overlapped pairs $\mathbb{E}\left[K^p_{(\cdot)}\right]$.

We denote events $I_{\text{SNN}}^{(k)} = \{D_{ab,a\in\text{AR}^{(k)}(d),b\in\text{AC}^{(k)}(d)} = D_{aj,a\in\text{AR}^{(k)}(d)} = D_{ib,b\in\text{AC}^{(k)}(d)} = d\}$, and $I_{\text{MSNN}}^{(k)} = \{D_{ab,a\in\text{MAR}^{(k)}(d),b\in\text{MAC}^{(k)}(d)} = D_{ib,b\in\text{MAC}^{(k)}(d)}, D_{aj,a\in\text{MAR}^{(k)}(d)} = d\}$, given group $(k)$ and corresponding row/column sets $(\text{M})\text{AR}^{(k)}(d)$ and $(\text{M})\text{AC}^{(k)}(d)$. The number of overlapped rows $\left|(\text{M})\text{AR}^{(1)}(d)\cap(\text{M})\text{AR}^{(2)}(d)\right|$ can vary from 1 to $r$, while the number of overlapped columns can vary from 0 to $c$ (despite the case of the two matrices are the same).

For notation simplicity, we denote $\beta(r',c') := \binom{m-1-r}{r'}\binom{r}{r'}\binom{n-1-c}{c'}\binom{c}{c'}$. By taking expectation,

$$\mathbb{E}\left[K'_{\text{SNN}}\right] - \mathbb{E}\left[K_{\text{SNN}}\right] \leq \mathbb{E}\left[K^p_{\text{SNN}}\right]$$

$$=\mathbb{E}\left[\sum_{\substack{\text{AR}^{(1,2)}(d)\subseteq[m]\setminus i,\\ \text{AC}^{(1,2)}(d)\subseteq[n]\setminus j,\\ \{\text{AR}^{(1)}(d),\text{AC}^{(1)}(d)\}\neq\{\text{AR}^{(2)}(d),\text{AC}^{(2)}(d)\}}} \mathbf{1}\left\{I_{\text{SNN}}^{(1)}, I_{\text{SNN}}^{(2)}\right\}\right]$$

$$=\sum_{\substack{\text{AR}^{(1,2)}(d)\subseteq[m]\setminus i,\\ \text{AC}^{(1,2)}(d)\subseteq[n]\setminus j,\\ \{\text{AR}^{(1)}(d),\text{AC}^{(1)}(d)\}\neq\{\text{AR}^{(2)}(d),\text{AC}^{(2)}(d)\}}} \mathbb{E}\left[\mathbf{1}\left\{I_{\text{SNN}}^{(1)}, I_{\text{SNN}}^{(2)}\right\}\right] \tag{16}$$

$$=\frac{1}{2}\sum_{\substack{0\leq r'\leq r-1,\\ 0\leq c'\leq c,\\ (r',c')\neq(0,0)}} \binom{m-1}{r}\binom{n-1}{c}\beta(r',c')\cdot p_d^{r+r'} p_d^{c+c'} p_d^{2rc-(r-r')(c-c')},$$

Similarly,

$$\mathbb{E}\left[K'_{\text{MSNN}}\right] - \mathbb{E}\left[K_{\text{MSNN}}\right] \leq \mathbb{E}\left[K^p_{\text{MSNN}}\right]$$

$$=\mathbb{E}\left[\sum_{\substack{\text{MAR}^{(1,2)}(d)\subseteq[m]\setminus i,\\ \text{MAC}^{(1,2)}(d)\subseteq[n]\setminus j,\\ \{\text{MAR}^{(1)}(d),\text{MAC}^{(1)}(d)\}\neq\{\text{MAR}^{(2)}(d),\text{MAC}^{(2)}(d)\}}} \mathbf{1}\left\{I^{(1)}_{\text{MSNN}}, I^{(2)}_{\text{MSNN}}\right\}\right]$$

$$=\sum_{\substack{\text{MAR}^{(1,2)}(d)\subseteq[m]\setminus i,\\ \text{MAC}^{(1,2)}(d)\subseteq[n]\setminus j,\\ \{\text{MAR}^{(1)}(d),\text{MAC}^{(1)}(d)\}\neq\{\text{MAR}^{(2)}(d),\text{MAC}^{(2)}(d)\}}} \mathbb{E}\left[\mathbf{1}\left\{I^{(1)}_{\text{MSNN}}, I^{(2)}_{\text{MSNN}}\right\}\right] \tag{17}$$

$$=\frac{1}{2}\sum_{\substack{0\leq r'\leq r-1,\\ 0\leq c'\leq c,\\ (r',c')\neq(0,0)}} \binom{m-1}{r}\binom{n-1}{c}\beta(r',c')\cdot p_d^{r+r'}\left(\sum_{d'\in\mathcal{L}}p_{d'}^{r+1}\right)^{2c'}\left(\sum_{d'\in\mathcal{L}}p_{d'}^{r+r'+1}\right)^{c-c'}.$$

From properties of combinatorial number, $\beta(r',c') \leq \frac{(mr)^{r'}(nc)^{c'}}{(r')!^2(c')!^2}$.

From the property of the Modified Bessel Function of the First Kind, Order Zero, $I_0(x) \leq \frac{1}{1-x^2/4}$ for $0 \leq x < 2$.

By Lemma D.4 and condition (13), we have

$$2\mathbb{E}\left[K^p_{\text{SNN}}\right]/\mathbb{E}\left[K'_{\text{SNN}}\right]$$

$$=\sum_{\substack{0\leq r'\leq r-1,\\ 0\leq c'\leq c,\\ (r',c')\neq(0,0)}} \beta(r',c')\cdot p_d^{r'+c'+r'c+rc'-r'c'}$$

$$\leq \left(I_0\left(2\sqrt{mrp_d^{c+1}}\right)-1\right)+\left(I_0\left(2\sqrt{ncp_d^{r+1}}\right)-1\right) \tag{18}$$

$$+\left(I_0\left(2\sqrt{mrp_d^{\alpha c+1}}\right)-1\right)\left(I_0\left(2\sqrt{ncp_d^{r+1+\alpha(1-r)}}\right)-1\right)$$

$$=O\left(mrp_d^{c+1}+ncp_d^{r+1}+mrncp_d^{(1-\alpha)r+\alpha c+2+\alpha}\right),$$

and

$$2\mathbb{E}\left[K_{\text{MSNN}}^p\right]/\mathbb{E}\left[K_{\text{MSNN}}'\right]$$

$$= \sum_{\substack{0 \le r' \le r-1, \\ 0 \le c' \le c, \\ (r',c') \ne (0,0)}} \beta(r',c') \cdot p_d^{r'} \left(\sum_{d' \in \mathcal{L}} p_{d'}^{r+1}\right)^{2c'-c} \left(\sum_{d' \in \mathcal{L}} p_{d'}^{r+r'+1}\right)^{c-c'}$$

$$\le \sum_{\substack{0 \le r' \le r-1, \\ 0 \le c' \le c, \\ (r',c') \ne (0,0)}} \beta(r',c') \cdot p_d^{r'} \left(\sum_{d' \in \mathcal{L}} p_{d'}^{r+1}\right)^{c'} p_{\max}^{r'(c-c')} \tag{19}$$

$$\le \left(I_0\left(2\sqrt{mrp_d p_{\max}^c}\right) - 1\right) + \left(I_0\left(2\sqrt{nc\left(\sum_{d' \in \mathcal{L}} p_{d'}^{r+1}\right)}\right) - 1\right)$$

$$+ \left(I_0\left(2\sqrt{mrp_d p_{\max}^{\alpha c}}\right) - 1\right)\left(I_0\left(2\sqrt{nc\left(\sum_{d' \in \mathcal{L}} p_{d'}^{r+1}\right) p_{\max}^{\alpha(1-r)}}\right) - 1\right)$$

$$= O\left(mrp_d p_{\max}^c + nc\gamma p_{\max}^{r+1} + mrnc\gamma p_d p_{\max}^{\alpha c + (1-\alpha)r + 1 + \alpha}\right).$$

By combining the results above with Lemma D.3, under (13) we have

$$\mathbb{E}\left[K_{\text{SNN}}(d)\right] \in \left[\frac{1}{1 + O\left(mrp_d^{c+1} + ncp_d^{r+1} + mrncp_d^{(1-\alpha)r + \alpha c + 2 + \alpha}\right)}, 1\right]$$

$$\cdot \binom{m-1}{r}\binom{n-1}{c} p_d^{rc+r+c},$$

$$\mathbb{E}\left[K_{\text{MSNN}}(d)\right] \in \left[\frac{1}{1 + O\left(mrp_d p_{\max}^c + nc\gamma p_{\max}^{r+1} + mrnc\gamma p_d p_{\max}^{\alpha c + (1-\alpha)r + 1 + \alpha}\right)}, 1\right] \tag{20}$$

$$\cdot \binom{m-1}{r}\binom{n-1}{c} \gamma^c p_d^r p_{\max}^{(r+1)c}.$$

Further by the sparsity condition in Theorem 5.8,

$$O\left(mrp_d^{c+1} + ncp_d^{r+1} + mrncp_d^{(1-\alpha)r + \alpha c + 2 + \alpha}\right) = o(1),$$

$$O\left(mrp_d p_{\max}^c + nc\gamma p_{\max}^{r+1} + mrnc\gamma p_d p_{\max}^{\alpha c + (1-\alpha)r + 1 + \alpha}\right) = o(1). \tag{21}$$

This completes the proof.

$\square$

Proof of Corollary 5.10:

*Proof.* By Theorem 5.8 we have

$$\mathbb{E}\left[K_{\text{SNN}}(d)\right] = (1 - o(1))\binom{m-1}{r}\binom{n-1}{c} p_d^{rc+r+c},$$

$$\mathbb{E}\left[K_{\text{MSNN}}(d)\right] = (1 - o(1))\binom{m-1}{r}\binom{n-1}{c} \gamma^c p_d^r p_{\max}^{(r+1)c}. \tag{22}$$

By taking division we complete the proof.

$\square$

Proof of Corollary 5.11:

*Proof.* From Theorem 5.8,

$$\mathbb{E}\left[K_{\text{SNN}}(d)\right] = (1 - o(1))\binom{m-1}{r}\binom{n-1}{c}p_d^{rc+r+c},$$

$$\mathbb{E}\left[K_{\text{MSNN}}(d)\right] = (1 - o(1))\binom{m-1}{r}\binom{n-1}{c}\gamma^c p_d^r p_{\max}^{(r+1)c}, \tag{23}$$

$$\mathbb{E}\left[K_{\text{MSNN}}(d_{\max})\right] = (1 - o(1))\binom{m-1}{r}\binom{n-1}{c}\gamma^c p_{\max}^{rc+r+c}.$$

By taking division, we complete the proof.

$\square$

### D.3.3. DATA EFFICIENCY UNDER MISSING NOT AT RANDOM

This section presents the formal version and the proof of Corollary 5.13 in section 5.3.2.

Recall the MNAR probability distribution defined in Section 5.3.2:

$$p_{\text{MNAR}}\left(A_{ij}^{d' \in \mathcal{L}}, d\right) = \frac{e^{\lambda_d A_{ij}^d + \delta_d}}{s_0 + \sum_{d' \in \mathcal{L}} e^{\lambda_{d'} A_{ij}^{d'} + \delta_{d'}}}. \tag{24}$$

For simplicity, we define $p_{\max_{(i,j)},d} := \max_{(a,b) \in \mathbb{R}^m \times \mathbb{R}^n \setminus (i,j)} p_{\text{MNAR}}\left(A_{ab}^{d' \in \mathcal{L}}, d\right)$, $p_{\min_{(i,j)},d} := \min_{(a,b) \in \mathbb{R}^m \times \mathbb{R}^n \setminus (i,j)} p_{\text{MNAR}}\left(A_{ab}^{d' \in \mathcal{L}}, d\right)$.

**Theorem D.5.** *Expectation of number of samples under MNAR setting.*

*Consider estimation of any entry $(i,j)$ under treatment level $d$. Suppose the treatment assignment $D$ follows missing not at random distribution (3), for fixed sample size $\text{AR}(d) = \text{MAR}(d) = r$ and $\text{AC}(d) = \text{MAC}(d) = c$, if the probabilities are sufficiently small, that is for some $\alpha \in (0, 1)$ the following sparsity condition holds:*

$$mrp_{\max_{(i,j)},d}\left(\max_{d'} p_{\max_{(i,j)},d'}\right)^{\alpha c} = o\left(\left(\frac{p_{\max_{(i,j)},d}}{p_{\min_{(i,j)},d}}\right)^{-r}\left(\max\left(\frac{\sum_{d' \in \mathcal{L}} p_{\max_{(i,j)},d'}^{r+1}}{\sum_{d' \in \mathcal{L}} p_{\min_{(i,j)},d'}^{r+1}}, \frac{p_{\max_{(i,j)},d}^{r+1}}{p_{\min_{(i,j)},d}^{r+1}}\right)\right)^{-c}\right),$$

$$nc\left(\max_{d'} p_{\max_{(i,j)},d'}\right)^{\alpha(1-r)}\left(\sum_{d' \in \mathcal{L}} p_{\max_{(i,j)},d'}^{r+1}\right) = o\left(\left(\frac{p_{\max_{(i,j)},d}}{p_{\min_{(i,j)},d}}\right)^{-r}\left(\max\left(\frac{\sum_{d' \in \mathcal{L}} p_{\max_{(i,j)},d'}^{r+1}}{\sum_{d' \in \mathcal{L}} p_{\min_{(i,j)},d'}^{r+1}}, \frac{p_{\max_{(i,j)},d}^{r+1}}{p_{\min_{(i,j)},d}^{r+1}}\right)\right)^{-c}\right). \tag{25}$$

*then the expectations of $K_{(.)}$ are given by:*

$$\mathbb{E}\left[K_{\text{SNN}}\right]$$

$$=(1-o(1))\cdot\sum_{\substack{\text{AR}(d)\subseteq[m]\backslash i,\\ \text{AC}(d)\subseteq[n]\backslash j}}\left[\prod_{\substack{a\in\text{AR}(d)\cup\{i\},\\ b\in\text{AC}(d)}}p_{\text{MNAR}}\left(A_{ab}^{d'\in\mathcal{L}},d\right)\right]\left[\prod_{a\in\text{AR}(d)}p_{\text{MNAR}}\left(A_{aj}^{d'\in\mathcal{L}},d\right)\right],$$

$$\mathbb{E}\left[K_{\text{MSNN}}\right]$$

$$=(1-o(1))\cdot\sum_{\substack{\text{MAR}(d)\subseteq[m]\backslash i,\\ \text{MAC}(d)\subseteq[n]\backslash j}}\left\{\prod_{b\in\text{MAC}(d)}\sum_{d''\in\mathcal{L}}\left[\prod_{a\in\text{MAR}(d)\cup\{i\}}p_{\text{MNAR}}\left(A_{ab}^{d'\in\mathcal{L}},d''\right)\right]\right\}\left[\prod_{a\in\text{MAR}(d)}p_{\text{MNAR}}\left(A_{aj}^{d'\in\mathcal{L}},d\right)\right].$$

$$(26)$$

*Furthermore, the sample efficiency is lower&upper bounded by*

$$\mathbb{E}\left[K_{\text{MSNN}}\right]/\mathbb{E}\left[K_{\text{SNN}}\right]\geq(1-o(1))\cdot\left(1+\left(\sum_{d'\in\mathcal{L}\backslash\{d\}}e^{(r+1)\left(-|\lambda_{d'}|f(d')+\delta_{d'}\right)}\right)e^{(r+1)(-|\lambda_d|f(d)-\delta_d)}\right)^c,$$

$$\mathbb{E}\left[K_{\text{MSNN}}\right]/\mathbb{E}\left[K_{\text{SNN}}\right]\leq(1+o(1))\cdot\left(1+\left(\sum_{d'\in\mathcal{L}\backslash\{d\}}e^{(r+1)\left(|\lambda_{d'}|f(d')+\delta_{d'}\right)}\right)e^{(r+1)(|\lambda_d|f(d)-\delta_d)}\right)^c.$$

$$(27)$$

*Proof.* From definitions of $p_{\max_{(i,j)},d}$ and $p_{\min_{(i,j)},d}$,

$$
\begin{aligned}
p_{\max_{(i,j)},d} &= \max_{(a,b)\in\mathbb{R}^m\times\mathbb{R}^n\backslash(i,j)}\frac{e^{\lambda_d A_{ab}^d+\delta_d}}{s_0+\sum_{d'\in\mathcal{L}}e^{\lambda_{d'}A_{ab}^{d'}+\delta_{d'}}}\\
&=\max_{(a,b)\in\mathbb{R}^m\times\mathbb{R}^n\backslash(i,j)}\frac{1}{s_0 e^{-\left(\lambda_d A_{ab}^d+\delta_d\right)}+\sum_{d'\in\mathcal{L}}e^{\left(\lambda_{d'}A_{ab}^{d'}+\delta_{d'}\right)-\left(\lambda_d A_{ab}^d+\delta_d\right)}}\\
&=\frac{1}{\min_{(a,b)\in\mathbb{R}^m\times\mathbb{R}^n\backslash(i,j)}\left[s_0 e^{-\left(\lambda_d A_{ab}^d+\delta_d\right)}+\sum_{d'\in\mathcal{L}}e^{\left(\lambda_{d'}A_{ab}^{d'}+\delta_{d'}\right)-\left(\lambda_d A_{ab}^d+\delta_d\right)}\right]}\\
&\leq\frac{1}{1+\left[s_0+\sum_{d'\in\mathcal{L}\backslash\{d\}}e^{-|\lambda_{d'}|f(d')+\delta_{d'}}\right]e^{-|\lambda_d|f(d)-\delta_d}}.
\end{aligned}
$$

$$(28)$$

Similarly,

$$p_{\min_{(i,j)},d}\geq\frac{1}{1+\left[s_0+\sum_{d'\in\mathcal{L}\backslash\{d\}}e^{|\lambda_{d'}|f(d')+\delta_{d'}}\right]e^{|\lambda_d|f(d)-\delta_d}}.$$

$$(29)$$

For simplicity, we omit $(d)$ in $K_{(\cdot)}(d)$, $K'_{(\cdot)}(d)$ and $K^p_{(\cdot)}(d)$ during the proof.

Similar to the proof of MCAR, for the upper bound, we consider $\mathbb{E}\left[K'_{(\cdot)}\right]$:

$$\mathbb{E}\left[K_{\text{SNN}}\right] \leq \mathbb{E}\left[K'_{\text{SNN}}\right]$$

$$= \sum_{\substack{\text{AR}(d)\subseteq[m]\setminus i,\\ \text{AC}(d)\subseteq[n]\setminus j}} \mathbb{E}\left[\mathbf{1}\{D_{ab,a\in\text{AR}(d),b\in\text{AC}(d)} = D_{aj,a\in\text{AR}(d)} = D_{ib,b\in\text{AC}(d)} = d\}\right]$$

$$= \sum_{\substack{\text{AR}(d)\subseteq[m]\setminus i,\\ \text{AC}(d)\subseteq[n]\setminus j}} \left[\prod_{\substack{a\in\text{AR}(d)\cup\{i\},\\ b\in\text{AC}(d)}} p_{\text{MNAR}}\left(A_{ab}^{d'\in\mathcal{L}}, d\right)\right]\left[\prod_{a\in\text{AR}(d)} p_{\text{MNAR}}\left(A_{aj}^{d'\in\mathcal{L}}, d\right)\right], \tag{30}$$

$$\mathbb{E}\left[K_{\text{MSNN}}\right] \leq \mathbb{E}\left[K'_{\text{MSNN}}\right]$$

$$= \sum_{\substack{\text{MAR}(d)\subseteq[m]\setminus i,\\ \text{MAC}(d)\subseteq[n]\setminus j}} \mathbb{E}\left[\mathbf{1}\{D_{ab,a\in\text{MAR}(d),b\in\text{MAC}(d)} = D_{ib,b\in\text{MAC}(d)}, D_{aj,a\in\text{MAR}(d)} = d\}\right]$$

$$= \sum_{\substack{\text{MAR}(d)\subseteq[m]\setminus i,\\ \text{MAC}(d)\subseteq[n]\setminus j}} \left\{\prod_{b\in\text{MAC}(d)} \sum_{d''\in\mathcal{L}} \left[\prod_{a\in\text{MAR}(d)\cup\{i\}} p_{\text{MNAR}}\left(A_{ab}^{d'\in\mathcal{L}}, d''\right)\right]\right\}\left[\prod_{a\in\text{MAR}(d)} p_{\text{MNAR}}\left(A_{aj}^{d'\in\mathcal{L}}, d\right)\right]. \tag{31}$$

Thus

$$\mathbb{E}\left[K'_{\text{SNN}}\right] \geq \binom{m-1}{r}\binom{n-1}{c}p_{\min_{(i,j)},d}^{rc+r+c},$$

$$\mathbb{E}\left[K'_{\text{MSNN}}\right] \geq \binom{m-1}{r}\binom{n-1}{c}\left[p_{\min_{(i,j)},d}^{r}\cdot\left(\sum_{d'\in\mathcal{L}} p_{\min_{(i,j)},d'}^{r+1}\right)^{c}\right], \tag{32}$$

$$\mathbb{E}\left[K'_{\text{MSNN}}\right]$$

$$= \sum_{\substack{\text{MAR}(d)\subseteq[m]\setminus i,\\ \text{MAC}(d)\subseteq[n]\setminus j}} \left\{\prod_{b\in\text{MAC}(d)} \left\{\left[\prod_{a\in\text{MAR}(d)\cup\{i\}} p_{\text{MNAR}}\left(A_{ab}^{d'\in\mathcal{L}}, d\right)\right]\left[\sum_{d''\in\mathcal{L}} \frac{\prod_{a\in\text{MAR}(d)\cup\{i\}} p_{\text{MNAR}}\left(A_{ab}^{d'\in\mathcal{L}}, d''\right)}{\prod_{a\in\text{MAR}(d)\cup\{i\}} p_{\text{MNAR}}\left(A_{ab}^{d'\in\mathcal{L}}, d\right)}\right]\right\}\right\}$$

$$\cdot\left[\prod_{a\in\text{MAR}(d)} p_{\text{MNAR}}\left(A_{aj}^{d'\in\mathcal{L}}, d\right)\right]$$

$$= \sum_{\substack{\text{MAR}(d)\subseteq[m]\setminus i,\\ \text{MAC}(d)\subseteq[n]\setminus j}} \left\{\prod_{b\in\text{MAC}(d)} \left\{\left[\prod_{a\in\text{MAR}(d)\cup\{i\}} p_{\text{MNAR}}\left(A_{ab}^{d'\in\mathcal{L}}, d\right)\right]\left[\sum_{d''\in\mathcal{L}} \prod_{a\in\text{MAR}(d)\cup\{i\}} \frac{p_{\text{MNAR}}\left(A_{ab}^{d'\in\mathcal{L}}, d''\right)}{p_{\text{MNAR}}\left(A_{ab}^{d'\in\mathcal{L}}, d\right)}\right]\right\}\right\}$$

$$\cdot\left[\prod_{a\in\text{MAR}(d)} p_{\text{MNAR}}\left(A_{aj}^{d'\in\mathcal{L}}, d\right)\right]$$

$$\in \left[\left(1 + \left(\sum_{d''\in\mathcal{L}\setminus\{d\}} e^{(r+1)\left(-|\lambda_{d''}|f(d'')+\delta_{d''}\right)}\right)e^{(r+1)\left(-|\lambda_d|f(d)-\delta_d\right)}\right)^{c},\right.$$

$$\left.\left(1 + \left(\sum_{d''\in\mathcal{L}\setminus\{d\}} e^{(r+1)\left(|\lambda_{d''}|f(d'')+\delta_{d''}\right)}\right)e^{(r+1)\left(|\lambda_d|f(d)-\delta_d\right)}\right)^{c}\right]\cdot\mathbb{E}\left[K'_{\text{SNN}}\right] \tag{33}$$

The last step of equation (33) is from the fact that $\forall\text{MAR}(d) \subseteq [m]\setminus i$ such that $|\text{MAR}(d)| = r$, $\forall b \in [n]\setminus j$,

$$\sum_{d''\in\mathcal{L}}\prod_{a\in\text{MAR}(d)\cup\{i\}}\frac{p_{\text{MNAR}}\left(A_{ab}^{d'\in\mathcal{L}},d''\right)}{p_{\text{MNAR}}\left(A_{ab}^{d'\in\mathcal{L}},d\right)}$$

$$=1+\sum_{d''\in\mathcal{L}\backslash\{d\}}\prod_{a\in\text{MAR}(d)\cup\{i\}}\frac{p_{\text{MNAR}}\left(A_{ab}^{d'\in\mathcal{L}},d''\right)}{p_{\text{MNAR}}\left(A_{ab}^{d'\in\mathcal{L}},d\right)}$$

$$=1+\sum_{d''\in\mathcal{L}\backslash\{d\}}\prod_{a\in\text{MAR}(d)\cup\{i\}}\frac{e^{\lambda_{d''}A_{ab}^{d''}+\delta_{d''}}}{e^{\lambda_d A_{ab}^d+\delta_d}}$$

$$\in\left[1+\left(\sum_{d''\in\mathcal{L}\backslash\{d\}}e^{(r+1)\left(-|\lambda_{d''}|f(d'')+\delta_{d''}\right)}\right)e^{(r+1)(-|\lambda_d|f(d)-\delta_d)},1+\left(\sum_{d''\in\mathcal{L}\backslash\{d\}}e^{(r+1)\left(|\lambda_{d''}|f(d'')+\delta_{d''}\right)}\right)e^{(r+1)(|\lambda_d|f(d)-\delta_d)}\right].$$
(34)

Then we upper bound the number of overlapped pairs $\mathbb{E}\left[K_{(\cdot)}^p\right]$.

Following the proof strategy of MCAR, we also denote events $I_{\text{SNN}}^{(k)}=\{D_{ab,a\in\text{AR}^{(k)}(d),b\in\text{AC}^{(k)}(d)}=D_{aj,a\in\text{AR}^{(k)}(d)}=D_{ib,b\in\text{AC}^{(k)}(d)}=d\}$, and $I_{\text{MSNN}}^{(k)}=\{D_{ab,a\in\text{MAR}^{(k)}(d),b\in\text{MAC}^{(k)}(d)}=D_{ib,b\in\text{MAC}^{(k)}(d)},D_{aj,a\in\text{MAR}^{(k)}(d)}=d\}$, given group $(k)$ and corresponding row/column sets $(\text{M})\text{AR}^{(k)}(d)$ and $(\text{M})\text{AC}^{(k)}(d)$. Denote $\beta(r',c'):=\binom{m-1-r}{r'}\binom{r}{r'}\binom{n-1-c}{c'}\binom{c}{c'}$. By taking expectation,

$$\mathbb{E}\left[K_{\text{SNN}}^p\right]=\sum_{\substack{\text{AR}^{(1,2)}(d)\subseteq[m]\backslash i,\\\text{AC}^{(1,2)}(d)\subseteq[n]\backslash j,\\\{\text{AR}^{(1)}(d),\text{AC}^{(1)}(d)\}\neq\{\text{AR}^{(2)}(d),\text{AC}^{(2)}(d)\}}}\mathbb{E}\left[\mathbf{1}\left\{I_{\text{SNN}}^{(1)},I_{\text{SNN}}^{(2)}\right\}\right]$$

$$\leq\frac{1}{2}\sum_{\substack{0\leq r'\leq r-1,\\0\leq c'\leq c,\\(r',c')\neq(0,0)}}\binom{m-1}{r}\binom{n-1}{c}\beta(r',c')\cdot p_{\max_{(i,j)},d}^{r+r'}p_{\max_{(i,j)},d}^{c+c'}p_{\max_{(i,j)},d}^{2rc-(r-r')(c-c')},$$
(35)

$$\mathbb{E}\left[K_{\text{MSNN}}^p\right]=\sum_{\substack{\text{MAR}^{(1,2)}(d)\subseteq[m]\backslash i,\\\text{MAC}^{(1,2)}(d)\subseteq[n]\backslash j,\\\{\text{MAR}^{(1)}(d),\text{MAC}^{(1)}(d)\}\neq\{\text{MAR}^{(2)}(d),\text{MAC}^{(2)}(d)\}}}\mathbb{E}\left[\mathbf{1}\left\{I_{\text{MSNN}}^{(1)},I_{\text{MSNN}}^{(2)}\right\}\right]$$

$$\leq\frac{1}{2}\sum_{\substack{0\leq r'\leq r-1,\\0\leq c'\leq c,\\(r',c')\neq(0,0)}}\binom{m-1}{r}\binom{n-1}{c}\beta(r',c')\cdot p_{\max_{(i,j)},d}^{r+r'}\left(\sum_{d'\in\mathcal{L}}p_{\max_{(i,j)},d'}^{r+1}\right)^{2c'}\left(\sum_{d'\in\mathcal{L}}p_{\max_{(i,j)},d'}^{r+r'+1}\right)^{c-c'}.$$
(36)

Then by Lemma D.4 and condition (25),

$$2\mathbb{E}\left[K_{\mathrm{SNN}}^{p}\right]/\mathbb{E}\left[K_{\mathrm{SNN}}'\right]$$

$$\leq \left(\frac{p_{\max_{(i,j)},d}}{p_{\min_{(i,j)},d}}\right)^{r+c+rc}\left[\sum_{\substack{0\leq r'\leq r-1,\\0\leq c'\leq c,\\(r',c')\neq(0,0)}}\beta(r',c')\cdot p_{\max_{(i,j)},d}^{r'+c'+r'c+rc'-r'c'}\right] \tag{37}$$

$$=O\left(\left(\frac{p_{\max_{(i,j)},d}}{p_{\min_{(i,j)},d}}\right)^{r+c+rc}\left(mrp_{\max_{(i,j)},d}^{c+1}+ncp_{\max_{(i,j)},d}^{r+1}+mrncp_{\max_{(i,j)},d}^{(1-\alpha)r+\alpha c+2+\alpha}\right)\right)$$

$$=o(1),$$

$$2\mathbb{E}\left[K_{\mathrm{MSNN}}^{p}\right]/\mathbb{E}\left[K_{\mathrm{MSNN}}'\right]$$

$$\leq \left(\frac{p_{\max_{(i,j)},d}}{p_{\min_{(i,j)},d}}\right)^{r}\left(\frac{\sum_{d'\in\mathcal{L}}p_{\max_{(i,j)},d'}^{r+1}}{\sum_{d'\in\mathcal{L}}p_{\min_{(i,j)},d'}^{r+1}}\right)^{c}$$

$$\cdot\left[\sum_{\substack{0\leq r'\leq r-1,\\0\leq c'\leq c,\\(r',c')\neq(0,0)}}\beta(r',c')\cdot p_{\max_{(i,j)},d}^{r'}\left(\sum_{d'\in\mathcal{L}}p_{\max_{(i,j)},d'}^{r+1}\right)^{2c'-c}\left(\sum_{d'\in\mathcal{L}}p_{\max_{(i,j)},d'}^{r+r'+1}\right)^{c-c'}\right] \tag{38}$$

$$=O\left(\left(\frac{p_{\max_{(i,j)},d}}{p_{\min_{(i,j)},d}}\right)^{r}\left(\frac{\sum_{d'\in\mathcal{L}}p_{\max_{(i,j)},d'}^{r+1}}{\sum_{d'\in\mathcal{L}}p_{\min_{(i,j)},d'}^{r+1}}\right)^{c}\cdot\left[mrp_{\max_{(i,j)},d}\left(\max_{d'}p_{\max_{(i,j)},d'}\right)^{c}\right.\right.$$

$$\left.\left.+nc\left(\sum_{d'\in\mathcal{L}}p_{\max_{(i,j)},d'}^{r+1}\right)+mrncp_{\max_{(i,j)},d}\left(\max_{d'}p_{\max_{(i,j)},d'}\right)^{\alpha(c-r+1)}\left(\sum_{d'\in\mathcal{L}}p_{\max_{(i,j)},d'}^{r+1}\right)\right]\right)$$

$$=o(1).$$

Further by Lemma D.3,

$$\mathbb{E}\left[K_{\mathrm{SNN}}\right]$$

$$=(1-o(1))\cdot\sum_{\substack{\mathrm{AR}(d)\subseteq[m]\setminus i,\\\mathrm{AC}(d)\subseteq[n]\setminus j}}\left[\prod_{\substack{a\in\mathrm{AR}(d)\cup\{i\},\\b\in\mathrm{AC}(d)}}p_{\mathrm{MNAR}}\left(A_{ab}^{d'\in\mathcal{L}},d\right)\right]\left[\prod_{a\in\mathrm{AR}(d)}p_{\mathrm{MNAR}}\left(A_{aj}^{d'\in\mathcal{L}},d\right)\right],$$

$$\mathbb{E}\left[K_{\mathrm{MSNN}}\right]$$

$$=(1-o(1))\cdot\sum_{\substack{\mathrm{MAR}(d)\subseteq[m]\setminus i,\\\mathrm{MAC}(d)\subseteq[n]\setminus j}}\left\{\prod_{b\in\mathrm{MAC}(d)}\sum_{d''\in\mathcal{L}}\left[\prod_{a\in\mathrm{MAR}(d)\cup\{i\}}p_{\mathrm{MNAR}}\left(A_{ab}^{d'\in\mathcal{L}},d''\right)\right]\right\}\left[\prod_{a\in\mathrm{MAR}(d)}p_{\mathrm{MNAR}}\left(A_{aj}^{d'\in\mathcal{L}},d\right)\right]. \tag{39}$$

From (33), the proof is completed.

$$\square$$

### D.3.4. PROOFS OF KEY SUPPORTING LEMMAS

This section provides the proofs of key lemmas in Section D.3.1.

Proof of Lemma D.3:

*Proof.* For simplicity, we omit $(d)$ in $K_{(\cdot)}(d)$, $K'_{(\cdot)}(d)$ and $K^p_{(\cdot)}(d)$ during the proof.

The upper bound is constructed by omitting the row-disjoint constraint: $K_{(\cdot)} \leq K'_{(\cdot)}$. By taking expectation for both sides the upper bound is proven.

For the lower bound, we formalize the original problem into a graph problem with randomness, then apply the Caro-Wei Theorem (Caro, 1979; Wei, 1981).

Denote the index set of all possible $(\mathrm{M})\mathrm{AR}(d) \times (\mathrm{M})\mathrm{AC}(d)$ as $\mathcal{S}$ ($|\mathcal{S}| = \binom{m-1}{r}\binom{n-1}{c}$). Define the edge set $E_{\mathcal{S}} := \left\{ (k_1, k_2) : k_{1,2} \in \mathcal{S}, (\mathrm{M})\mathrm{AR}^{(k_1)}(d) \cap (\mathrm{M})\mathrm{AR}^{(k_2)}(d) \neq \emptyset \right\}$, the base graph $\mathcal{H} := (\mathcal{S}, E_{\mathcal{S}})$.

To address the randomness of activation, for index $k \in \mathcal{S}$ define $X_k = \mathbf{1}\{(\mathrm{M})\mathrm{AR}^{(k)}(d) \times (\mathrm{M})\mathrm{AC}^{(k)}(d) \text{ is valid}\}$ to be its indicator function. Specifically, for SNN $X_k = \mathbf{1}\{D_{ab:a\in\mathrm{AR}^{(k)}(d),b\in\mathrm{AC}^{(k)}(d)} = D_{aj:a\in\mathrm{AR}^{(k)}(d)} = D_{ib:b\in\mathrm{AC}^{(k)}(d)} = d\}$, for MSNN $X_k = \mathbf{1}\{D_{ab:a\in\mathrm{MAR}^{(k)}(d),b\in\mathrm{MAC}^{(k)}(d)} = D_{ib:b\in\mathrm{MAC}^{(k)}(d)}, D_{aj:a\in\mathrm{MAR}^{(k)}(d)} = d\}$. Thus $X_k = 1$ means subgroup $k$ can be a valid sample.

Under a given treatment assignment $\boldsymbol{D}$, we define the induced subgraph $\mathcal{G}(\boldsymbol{D})$ of $\{\mathcal{S}, E_{\mathcal{S}}\}$ on the active vertex set $V(\boldsymbol{D}) := \{k \in \mathcal{S} : X_k = 1\}$. Its edge set is naturally $E_{\mathcal{S}}(\boldsymbol{D}) = E_{\mathcal{S}} \cap (V(\boldsymbol{D}) \times V(\boldsymbol{D}))$. Then $K_{(\cdot)}$ is the independence number (size of the largest independent set) of the induced subgraph $\mathcal{G} = (V(\boldsymbol{D}), E_{\mathcal{S}}(\boldsymbol{D}))$.

From the Caro-Wei Theorem,

$$K_{(\cdot)} \mid \boldsymbol{D} \geq \sum_{k_1 \in V(\boldsymbol{D})} \frac{1}{1 + \deg(k_1)} = \sum_{k_1 \in V(\boldsymbol{D})} \frac{1}{1 + \sum_{k_2 \in V(\boldsymbol{D})\setminus\{k_1\}} \mathbf{1}\{(k_1, k_2) \in E_{\mathcal{S}}(\boldsymbol{D})\}}, \tag{40}$$

where $\deg(k_1) := \sum_{k_2 \in V(\boldsymbol{D})\setminus\{k_1\}} \mathbf{1}\{(k_1, k_2) \in E_{\mathcal{S}}(\boldsymbol{D})\}$ is the degree of vertex $k_1$.

By taking expectation over $\boldsymbol{D}$,

$$
\begin{aligned}
\mathbb{E}\left[K_{(\cdot)}\right] &\geq \mathbb{E}\left[\sum_{k_1 \in V(\boldsymbol{D})} \frac{1}{1 + \sum_{k_2 \in V(\boldsymbol{D})\setminus\{k_1\}} \mathbf{1}\{(k_1, k_2) \in E_{\mathcal{S}}(\boldsymbol{D})\}}\right] \\
&= \mathbb{E}\left[\sum_{k_1 \in \mathcal{S}} \frac{X_{k_1}}{1 + \sum_{k_2 \in \mathcal{S}\setminus\{k_1\}} \mathbf{1}\{(k_1, k_2) \in E_{\mathcal{S}}\} X_{k_1} X_{k_2}}\right] \\
&= \sum_{k_1 \in \mathcal{S}} \mathbb{E}\left[\frac{X_{k_1}}{1 + \sum_{k_2 \in \mathcal{S}\setminus\{k_1\}} \mathbf{1}\{(k_1, k_2) \in E_{\mathcal{S}}\} X_{k_1} X_{k_2}}\right] \\
&= \sum_{k_1 \in \mathcal{S}} \mathbb{P}(X_{k_1} = 1) \cdot \mathbb{E}\left[\frac{1}{1 + \sum_{k_2 \in \mathcal{S}\setminus\{k_1\}} \mathbf{1}\{(k_1, k_2) \in E_{\mathcal{S}}\} X_{k_2}} \Bigg| X_{k_1} = 1\right].
\end{aligned}
\tag{41}
$$

By applying Jensen's inequality,

$$
\begin{aligned}
\mathbb{E}\left[K_{(\cdot)}\right] &\geq \sum_{k_1 \in \mathcal{S}} \frac{\mathbb{P}(X_{k_1} = 1)}{\mathbb{E}\left[1 + \sum_{k_2 \in \mathcal{S}\setminus\{k_1\}} \mathbf{1}\{(k_1, k_2) \in E_{\mathcal{S}}\} X_{k_2} \Big| X_{k_1} = 1\right]} \\
&= \sum_{k_1 \in \mathcal{S}} \frac{\mathbb{P}(X_{k_1} = 1)}{1 + \sum_{k_2 \in \mathcal{S}\setminus\{k_1\}} \mathbf{1}\{(k_1, k_2) \in E_{\mathcal{S}}\} \mathbb{P}[X_{k_2} = 1 | X_{k_1} = 1]} \\
&= \sum_{k_1 \in \mathcal{S}} \frac{\mathbb{P}(X_{k_1})^2}{\mathbb{P}(X_{k_1}) + \sum_{\substack{k_2 \in \mathcal{S}\setminus\{k_1\}, \\ (k_1, k_2) \in E_{\mathcal{S}}}} \mathbb{P}\left(X_{k_1} \cap X_{k_2}\right)}.
\end{aligned}
\tag{42}
$$

By Titu's Lemma (or the Engel's Form of Cauchy-Schwarz inequality), for positive $a_k, b_k$, $\sum_k \frac{a_k^2}{b_k} \geq \frac{(\sum_k a_k)^2}{\sum_k b_k}$. This gives

$$\mathbb{E}\left[K_{(\cdot)}\right] \geq \frac{\left(\sum_{k_1 \in \mathcal{S}} \mathbb{P}(X_{k_1})\right)^2}{\sum_{k_1 \in \mathcal{S}} \mathbb{P}(X_{k_1}) + \sum_{\substack{k_{1,2} \in \mathcal{S}, \\ (k_1,k_2) \in E_{\mathcal{S}}}} \mathbb{P}\left(X_{k_1} \cap X_{k_2}\right)} = \frac{\mathbb{E}\left[K'_{(\cdot)}\right]^2}{\mathbb{E}\left[K'_{(\cdot)}\right] + 2\mathbb{E}\left[K^p_{(\cdot)}\right]}. \tag{43}$$

The last step is from the definition of $K^p_{(\cdot)}$. Then the proof for both sides are completed.

$\square$

Proof of Lemma D.4:

*Proof.* By reorganizing the summation,

$$\sum_{\substack{0 \leq r' \leq r-1, \\ 0 \leq c' \leq c, \\ (r',c') \neq (0,0)}} \binom{M}{r'}\binom{r}{r'}\binom{N}{c'}\binom{c}{c'} p_1^{r'} p_2^{(r+1)c'} p_3^{r'(c-c')}$$

$$= \sum_{1 \leq r' \leq r-1} \binom{M}{r'}\binom{r}{r'}(p_1 p_3^c)^{r'} + \sum_{1 \leq c' \leq c} \binom{N}{c'}\binom{c}{c'} p_2^{(r+1)c'} \tag{44}$$

$$+ \sum_{\substack{1 \leq r' \leq r-1, \\ 1 \leq c' \leq c}} \binom{M}{r'}\binom{r}{r'}\binom{N}{c'}\binom{c}{c'} p_1^{r'} p_2^{(r+1)c'} p_3^{r'(c-c')}.$$

By applying $\binom{x}{k} \leq \frac{x^k}{k!}$,

$$\sum_{r'=1}^{r-1} \binom{M}{r'}\binom{r}{r'}(p_1 p_3^c)^{r'} \leq \sum_{r'=1}^{r-1} \frac{(Mr p_1 p_3^c)^{r'}}{(r'!)^2} \leq \sum_{r'=1}^{+\infty} \frac{(Mr p_1 p_3^c)^{r'}}{(r'!)^2} \leq I_0\left(2\sqrt{Mr p_1 p_3^c}\right) - 1$$

$$\sum_{c'=1}^{c} \binom{N}{c'}\binom{c}{c'} p_2^{(r+1)c'} \leq \sum_{c'=1}^{c} \frac{(Nc p_2^{r+1})^{c'}}{(c'!)^2} \leq \sum_{c'=1}^{+\infty} \frac{(Nc p_2^{r+1})^{c'}}{(c'!)^2} \leq I_0\left(2\sqrt{Nc p_2^{r+1}}\right) - 1. \tag{45}$$

For the cross term, by appropriate scaling,

$$\sum_{\substack{1 \leq r' \leq r-1, \\ 1 \leq c' \leq c}} \binom{M}{r'}\binom{r}{r'}\binom{N}{c'}\binom{c}{c'} p_1^{r'} p_2^{(r+1)c'} p_3^{r'(c-c')}$$

$$= \sum_{\substack{1 \leq r' \leq r-1, \\ 1 \leq c' \leq c}} \binom{M}{r'}\binom{r}{r'}\binom{N}{c'}\binom{c}{c'} (p_1 p_3^{\alpha c})^{r'} \left(p_2^{r+1} p_3^{\alpha(1-r)}\right)^{c'} p_3^{\alpha(r-r'-1)c'+(1-\alpha)r'(c-c')}$$

$$\leq \sum_{\substack{1 \leq r' \leq r-1, \\ 1 \leq c' \leq c}} \binom{M}{r'}\binom{r}{r'}\binom{N}{c'}\binom{c}{c'} (p_1 p_3^{\alpha c})^{r'} \left(p_2^{r+1} p_3^{\alpha(1-r)}\right)^{c'} \tag{46}$$

$$= \left[\sum_{r'=1}^{r-1} \binom{M}{r'}\binom{r}{r'}(p_1 p_3^{\alpha c})^{r'}\right]\left[\sum_{c'=1}^{c} \binom{N}{c'}\binom{c}{c'}\left(p_2^{r+1} p_3^{\alpha(1-r)}\right)^{c'}\right]$$

$$\leq \left(I_0\left(2\sqrt{Mr p_1 p_3^{\alpha c}}\right) - 1\right)\left(I_0\left(2\sqrt{Nc p_2^{r+1} p_3^{\alpha(1-r)}}\right) - 1\right).$$

By combining result above, the proof is completed.

$\square$

