# OpenReview forum: "Causal Matrix Completion under Multiple Treatments via Mixed Synthetic Nearest Neighbors"
_ICML.cc/2026/Conference — ICML 2026 regular_

### Official Review · Reviewer_RVA3 · 2026-03-01

**Soundness:** 4
**Presentation:** 3
**Significance:** 2
**Originality:** 3
**Overall Recommendation:** 4
**Confidence:** 5

**Summary:**

Synthetic Nearest Neighbors (SNN), one of the earliest methods for matrix completion with theoretical guarantees under challenging conditions of MNAR, uses anchor rows and columns for estimation, and its success relies on sufficient data availability under each treatment level. (Aggarwal et. al. 2021) SNN demonstrated its success under binary treatment setups, but authors clearly point the limitations of SNN under multi-treatment setups. As a result, authors propose MSNN which estimates underlying true effect of a treatment level by leveraging observations across different treatment levels, and quantifies its improvements over SNN both theoretically and numerically.

**Compliance With Llm Reviewing Policy:**

Affirmed.

**Final Justification:**

Authors have answered all the initial questions I had, although most of which were lack of research regarding rigor of prior work on MC. This MSNN paper extends causal matrix completion to a more practical scenario of MC, the multi-treatment regime, which is the main reason behind my score of 4. As far as theoretical or technical novelty goes, the thm 1 and 2 of the paper are easy extensions of the equivalent theorems in SNN, and theorem 3 has minimal novel proof techniques. Despite that, I see the practical impact of MSNN and would maintain a score of 4.

**Key Questions For Authors:**

1. Assumption 2.5 is essential for theory. It would be great to include any discussion or remark regarding this assumption in the prior literature, as there are multiple examples in real-life applications where this asumption can be violated. If authors can't find such, it should be mentioned that this first time in literature such assumption is made. Additionally, authors can look into experiments checking the robustness of the MSNN against small deviations from assumption 2.5

**Limitations:**

Most of the limitations are already addressedin the significance section.

1. There has been multiple nearest neighbor approaches to MNAR matrix completion (some already mentioned before). It will be great to see how MSNN fares against other nearest neighbor based MC methods in the experiments section. Current experiment section is limited to MSNN vs SNN

Clarity:
Section headers can be more targeted and informative: for eg, instead of Section 2 header “Identification”, authors can write “Identification of multiple treatment levels”

Multiple typos, some were


”Anchor” rows in line 44


Subject to in line 276


AR(d) = MAR(d) = r and AC(d) = MAC(d) = c, |   | was missing.



I'll give a 4-Borderline accept for now. Proof of the theorems are fine (assuming lemmas are correct). MSNN has the potential to be a great paper, authors need to spend some time connecting it to existing works, writeup to improve the flow of readers and better position MSNN and its contribution in this field, and showing how does it perform against other nearest neighbor based MC methods.

**Strengths And Weaknesses:**

Soundness: The claims of improvement over SNN made by the authors are reasonably supported by their theorems and experiments. Although I didn't have time to check all the proofs of the theorems, most of the theoretical proofs followed the structure of (Aggarwal et. al. 2021) SNN paper, and the new assumption over SNN setup was more or less "expected" given their final goal.

Presentation and Significance: Although the main aim of MSNN was clearly portrayed, the authors only mention SNN in their introduction section. There have been multiple approaches to MNAR matrix completion/causal inference and authors can easily motivate their readers better by expanding their breadth of prior work covered, which will eventually highlight the contribution of MSNN to this sub-field. Pivotal examples of different approaches are [1] in JASA, which tackles MNAR matrix completion also using nearest neighbors under multiple treatments and a weak signal-to-noise ratio, [2] in AISTATS, which explores nearest neighbors for MNAR matrix completion without the need for construction of anchor rows and columns, and when low rank assumptions are violated, and empirical risk minimization (ERM) methods, precursor of the nearest neighbor approach for matrix completion based causal inference. These will better illustrate the hardness of problem MSNN is tackling to the readers. After first reading, it seemed that the current draft makes the contribution of MSNN "too specific" and merely an extension of SNN. (There was no hyperlink or mention of appendix section A in the main text where authors actually covered the rigor of prior work beyond SNN, readers are not expected to go through appendix sections which are not mentioned in main text)


[1] Matrix Completion When Missing Is Not at Random and Its Applications in Causal Panel Data Models, JASA 2025, Jungjun Choi, Ming Yuan


[2] On adaptivity and minimax optimality of two-sided nearest neighbors, AISTATS 2025, Tathagata Sadhukhan, Manit Paul, Raaz Dwivedi



The technical notation used in the intro section should be self-contained, and authors shouldn’t assume that all the readers are familiar with the specific notation used in the SNN paper, for example $\textit{“enabling the imputation coefficient $\beta$ to be estimated from a block of data that spans multiple treatment levels”}$. Introducing the notations properly before using them or better, explaining the concept in words in introduction section without using specific notation will improve the readability of the paper.

Originality: Out of the three main theorems, the first two theorems directly follow from theorems 2 and 3 of the SNN paper. So, only the third theorem was an original (and important in my opinion) theoretical contribution of the authors; therefore authors should specify which part(s) of that proof were most non-trivial or novel and not just a convex combination of older results. Moreover the writeup of the proof was hard to follow and then verify. A suggestion: just state all the lemmas required for proof first and then prove the main theorem first, and then prove the lemmas one by one.

$\textit{The most counterintuitive and theoretically impactful result
is the exponential improvement in sample efficiency for
sparse treatment levels.}$ Why is it counterintuitive? I might have missed it, but the premise and approach outlined in previous paragraphs seem to convey that there is indeed a big scope of improvement wrt sample efficiency in observational studies/RCTs with multiple treatment levels.

---

> ### Author Rebuttal · Authors · 2026-03-31
>
> We sincerely thank you for your expert and insightful comments. We have carefully considered your suggestions and greatly appreciate the opportunity to learn from them. We address your concerns point by point.
>
> **Presentation and Significance 1(lack of discussion of prior works)** Thank you for raising this point. In the revision, we have incorporated related works ([1,2]) you mentioned, other matrix completion works, and also connections to Counterfactual Regression and Causal Recommendation (see the response to W3&Q2 of reviewer dqF2). We have also revised and enriched introduction to compare our work with other methods, and have moved part of related works into main body, making better comparison and presenting the hardness of MSNN.
>
> **Presentation and Significance 2(notations in introduction)** Thank you for pointing this out. We have revised the expression to avoid heavy technologies in SNN: "The key innovation ..., enabling the imputation coefficients to be estimated from a block of data that spans \emph{multiple treatment levels}", "This is made possible..., which ensures the identifiability of the imputation coefficients across treatments ". This improves readability for readers not familiar with SNN.
>
> **Originality 1(theorem arrangement)** Thank you for this helpful suggestion. We have emphasized that the proofs of sample efficiency improvement under MCAR (and also MNAR, see the response to Weakness 1 of reviewer ``jnH3``) are the main novel contribution and are technically the most non-trivial, while the proofs of finite sample bound and asymptotic normality are reduction to SNN/following the outline of previous work. We have also rearranged the proof order in the appendix, stating the lemmas first and proving the main theorem then the lemmas.
>
> **Originality 2(why exponential increase is counterintuitive)** Classical intuition in causal inference and matrix completion (e.g., SNN ; Synthetic Interventions, Agarwal et al., 2020) suggests that estimation for a rare treatment fundamentally relies on sufficient within-treatment observations, and that moving to multiple treatments only worsens sample efficiency due to data fragmentation and the vanishing probability of valid anchors. Our result contradicts both intuitions. Under shared latent row factors (Assumption 2.5), MSNN enables cross-treatment identification of the imputation coefficients $\beta$, so that estimating $\beta$ no longer consumes scarce target-treatment samples. (Theorem 2.7, Algorithm 2) This converts the additional treatment dimension from a source of sparsity into a source of **combinatorial information**, allowing mixed anchors whose count scales exponentially in $c$. Such an exponential gain -- where sparsity induces combinatorial amplification rather than decay -- does not arise in standard nearest-neighbor or matrix completion frameworks.
>
> **Q1 (justification of core assumption)**
> Thank you for raising this important question. As noted in the paper, this assumption is adapted from Assumption 2 in [1], and is standard in low-rank causal models with structured treatments.
>
> The assumption states that each unit has intrinsic latent characteristics that remain invariant across treatments. In our model: $Y^{(d)} = U V^{(d)\top} + E^{(d)}$, the matrix $U$ captures stable unit-level traits, while $V^{(d)}$ captures treatment-specific effects.
>
> For example, in e-commerce platform, rows are users, treatments are discount levels, a user’s preference and price sensitivity remain stable, while different discounts affect outcomes via treatment-specific factors. Another example is the Proposition 99 case study in section 5.2, where rows are states and columns are years; this assumption means that the intrinsic factors that affect states' cigrette consumption remains stable despite fluctuation per year, thus the consumption behavior for one treatment can transfer to the other treatments.
>
> This assumption is essential for enabling **information sharing across treatments**. If latent row factors were treatment-dependent, observations from one treatment ``provides no information about another``, and the MSNN construction ``collapses to SNN``. By sharing latent row factor $U$, we can leverage information from data-rich levels to improve estimation in data-scarce levels. We have enriched discussion on the justification of this assumption, and have provided both theoretical analysis and **empirical sensitivity analysis** (also see the response to Weakness 3 of reviewer ``jnH3``, empirical result see Table https://anonymous.4open.science/r/ICML-rebuttal-MSNN-B460/jnH3/W3/Sensitivity_Table.png .
>
> [1] Agarwal, A., Shah, D., and Shen, D. Synthetic interventions. arXiv preprint arXiv:2006.07691, 2020.
>
> (continue: response window of ssQg)

---

> > ### Author Rebuttal · Reviewer_RVA3 · 2026-04-03
> >
> > Can the authors lay out the exact parameters for the simulation setup (like what is $n,m, r$, how many treatment levels are there, etc)? I believe it would give much more insight into the simulation findings.

---

> > > ### Author Response · Authors · 2026-04-04
> > >
> > > Thank you for your further question. The experimental details are provided in section B of Appendix, and the extra experiments we conducted during the rebuttal also follow the original hyperparamters. Specifically, we consider number of rows $m=300$, number of columns $n=100$, rank $r=3$, relative noise scale $\sigma = 0.001$, treatment levels includes: not observed, low, medium, high, very high, with MCAR treatment assignment scheme following:
> > >
> > > $
> > > p_{\rm MCAR}(d) = \begin{cases}
> > >     0.115 &, d = 0 \text{ (not observed)} \\\\
> > >     0.01 &, d = \text{low} \\\\
> > >     0.025 &, d = \text{medium} \\\\
> > >     0.05 &, d = \text{high} \\\\
> > >     0.8 &, d = \text{very high} \\\\
> > > \end{cases} .
> > > $
> > >
> > > This simulates the scenario where the low, medium, high treatments are of low probability to be observed, while a "very high" level has rich data. The scale for each treatment level ($f(d)$) is set by:
> > >
> > > $
> > > f(d) = \begin{cases}
> > >     1 &, d = \text{low} \\\\
> > >     5 &, d = \text{medium} \\\\
> > >     25 &, d = \text{high} \\\\
> > >     625 &, d = \text{very high}
> > > \end{cases} .
> > > $
> > >
> > > The selection of $f(d)$ does not affect the estimation result under MCAR, but only the probability distribution of which treatment level is observed for each entry under MNAR.
> > >
> > > Additionally, we have varied the noise scale $\sigma$ from $0.0001$ to $0.1$, as shown in Table https://anonymous.4open.science/r/ICML-rebuttal-MSNN-B460/sensitivity_noise_scale.png . This shows that the estimation error does not explode and steadily grows as the noise scale increases, giving that our estimation is robust against noise.
> > >
> > > Finally, thank you so much for further comments, and for your expert and insightful suggestions. We really appreciate your time and effort in reviewing this manuscript. For any further questions, we will respond as promptly as possible. Since anonymous links are only permitted on graphs and tables, we promise to release the codebase in the final version.

---

### Official Review · Reviewer_ssQg · 2026-03-13

**Soundness:** 3
**Presentation:** 2
**Significance:** 3
**Originality:** 3
**Overall Recommendation:** 5
**Confidence:** 4

**Summary:**

The success of SNN on entry wise identification and imputation in a matrix under MNAR relies on the construction of anchor rows and columns, which requires some amount of observation per treatment. For multiple treatment scenarios, the observations are normally sparse, which motivates this work. MSNN leverages information across the treatments, by exploiting the assumption of shared latent factors. For that end, mixed anchor rows and columns introduced. Theoretical and empirical results are thoroughly presented.

**Compliance With Llm Reviewing Policy:**

Affirmed.

**Final Justification:**

The paper has merit in generalizing SNN for multiple treatment scenarios, which is a practical one. The work directly addresses sparse observations and applies an appropriate assumption for leveraging information across the treatments. Theoretical and empirical results are thoroughly presented. Theoretical results are compared to that of SNN, and this gives a clear picture on the gains of MSNN. Further, the ambiguity of inference results were clarified. Overall, I will maintain my score.

**Key Questions For Authors:**

1. Can you clear the interpretation of the asymptotic results? What does it mean for normality to hold on the event E? Is E independent to sample size, meaning that it doesn't change as the estimator converge in distribution?

**Limitations:**

Many theoretical findings are put into the main text, and the math is difficult to read especially for fractions where the denominator contains combinatorial arguments. Only including the key findings of the theory and moving others to the appendix would have made the presentation cleaner.

**Strengths And Weaknesses:**

**Strengths**

- Clear motivation: Prevalence of multi-treatment setup (as in Synthetic Intervention) motivates the extension of SNN.

- Rigorous presentation and discussion of assumptions: Assumptions are clearly written and further justified with thorough discussion. Identification results are clearly stated.

- Comparison with SNN: Theoretical results are compared to that of SNN, this gives a clear picture on the gains of MSNN.

- Experiments on real data: The synthetic controls example is a widely used benchmark.


**Weaknesses**

- Ambiguity in inference results: The normality of estimator is on the event E. There seems to be a tradeoff: if sample size plays a role in event E, then the asymptotic normality result (on E, converge to Normal) is hard to interpret as the event itself is changing. If sample size doesn't play a role, then asymptotic result makes sense, but the event does not hold with probability that converge to 1.

---

> ### Author Rebuttal · Authors · 2026-03-31
>
> Thank you for your insightful questions and your recognition of our work. We address your concerns below separately.
>
> **W1&Q1 (ambiguity in event E on asymptotic results)**: Thank you for raising this insightful question. In Theorem 4.6, the event $E$ does depend on the sample size $m,n$. Formally, for some sequence $k \in \mathbb{N}_+$ and finite size event $E_k$ satisfying the assumptions in Theorem 4.6, the error sequence satisfies the asymptotic normality. For simplicity, we omit the index $k$ here in Theorem 4.6, following the expressions in Theorem 3 of the original SNN paper [1]. We have added a remark after this Theorem to clarify the interpretation of $E$.
>
> **Limitation (reading difficulty)**: Thank you for pointing out this point. We have rearranged the theorems and lemmas, and polished the math fomulas (for example, moving the combactorial numbers out of the fractions as you suggested).
>
> [1] Agarwal, Anish, et al. "Causal matrix completion." The thirty sixth annual conference on learning theory. PMLR, 2023.
>
> ---
> ---
> **(cont. ) Response to jnH3**
>
> We provide a theoretical explanation by omiting the error matrix $E^{(d)}$. For fixed $U^{(d)}$, $U^{(d^\prime)}$s are linear to $\zeta$. Now $x$ is also fixed, from Algorithm 2 $A_{ij}^{(d)} = q(\zeta = 0)^\top S(\zeta = 0)^+ x$. Since $q$ and $S$ are constructed by directly concatenating submatrices of $A^{(d^\prime)}(\zeta) = U^{(d^\prime)}(\zeta) V^{(d^\prime)\top}$ for $d^\prime \in \mathcal{L}$, $q$ and $S$ are linear to $\zeta$. Suppose $S(\zeta=0)$ is not ill-conditioned, by Talor's expansion the deviation $q^\top(\zeta) S(\zeta)^+ x - A_{ij}^{(d)}$ is also ``linear to`` $\zeta$ ``to the first order``.
>
> We have conducted the following empirical robustness analysis, where $\zeta$ varies from $1e-4$ to $1e-1$. Following similar settings in section 5.1, we compare the error on MCAR data in https://anonymous.4open.science/r/ICML-rebuttal-MSNN-B460/jnH3/W3/Sensitivity_Table.png. This shows that , the estimation error linearly grows with the noise level $\zeta$, supporting the theoretical analysis and showing robustness under moderate violation.
>
> **W4 (Missing Citations and Contextualization in the Introduction)**: Thanks. We have appropriately cited related works (e.g. SNN: [1]; MNAR in Matrix completion: [2]) for better and contextualization in the introduction as you suggested.
>
> [1] Agarwal, Anish, et al. "Causal matrix completion." The thirty sixth annual conference on learning theory. PMLR, 2023.
>
> [2] Ma, Wei, and George H. Chen. "Missing not at random in matrix completion: The effectiveness of estimating missingness probabilities under a low nuclear norm assumption." Advances in neural information processing systems 32 (2019).
>
> **W5 (inline equations)** Thank you for your suggestion. We have replaced the inline equations into centered equations for core assumptions and theorems.
>
> ---
> ---
> **(cont. ) Response to dqF2**
>
> **W5 (more baselines)** Thank you for your suggestion! We have added more baselines (CY [3], TS-NN [4]) in table https://anonymous.4open.science/r/ICML-rebuttal-MSNN-B460/RVA3/baselines/CY-TSNN.png, refering to the suggestion of reviewer RVA3 (Limitation 1). Both improve the SNN but still fail under the multiple treatment data, mostly because they are not particularly designed for estimation in data-scarse treatment level under multi-treatment.
>
> [3] Choi, Jungjun, and Ming Yuan. "Matrix completion when missing is not at random and its applications in causal panel data models." Journal of the American Statistical Association (2024): 1-15.
>
> [4] Sadhukhan, Tathagata, Manit Paul, and Raaz Dwivedi. "On adaptivity and minimax optimality of two-sided nearest neighbors." arXiv preprint arXiv:2411.12965 (2024).
>
> We are eager to know whether we have addressed your questions, and we would be happy to discuss further at any time to assist with your re-evaluation.
>
> ---
> ---
> **(cont. ) Response to RVA3**
>
> **Limitation 1 (More baselines)** Thank you for pointing out this part. In the revision, we have included additional baselines: CY algorithm from [2] and TS-NN from [3]. From the simulation results in https://anonymous.4open.science/r/ICML-rebuttal-MSNN-B460/RVA3/baselines/CY-TSNN.png, generally both cannot solve the data sparsity issue under the multiple treatment data (generally, they are not particularly designed for multi-treatment estimation), although achieves improvement towards SNN baseline.
>
> **Clarity and typos** Thank you very much for pointing these out. We have corrected all typos and improved section titles (e.g., “Identification of Multiple Treatment Effects”).
>
> [2] Choi, Jungjun, and Ming Yuan. "Matrix completion when missing is not at random and its applications in causal panel data models." Journal of the American Statistical Association (2024): 1-15.
>
> [3] Sadhukhan, Tathagata, Manit Paul, and Raaz Dwivedi. "On adaptivity and minimax optimality of two-sided nearest neighbors." arXiv preprint arXiv:2411.12965 (2024).

---

> > ### Author Rebuttal · Reviewer_ssQg · 2026-04-03
> >
> > The authors fully resolved the main question/concern I had. I will maintain my score 5 (accept).

---

> > > ### Author Response · Authors · 2026-04-04
> > >
> > > Thank you so much for your recognition of our work and for your thoughtful and constructive suggestions. We really appreciate your time and effort in reviewing this manuscript.

---

### Official Review · Reviewer_dqF2 · 2026-03-13

**Soundness:** 3
**Presentation:** 2
**Significance:** 2
**Originality:** 2
**Overall Recommendation:** 4
**Confidence:** 4

**Summary:**

This paper proposes a causal matrix completion method under missing-not-at-random with multiple treatments. Based on the assumption that all the treatment levels share the same latent row factors, the paper introduces a causal identification estimator by combining information across treatment levels. The theoretical properties of the method are analyzed, and experiments are conducted on a simulation and a case study to evaluate the performance of the proposed method.

**Compliance With Llm Reviewing Policy:**

Affirmed.

**Final Justification:**

I appreciate the responses of the authors. I think the submission is borderline but acceptable overall.

**Key Questions For Authors:**

1. What is the physical meaning of Assumption 2.5 in some specific application?

2. What is the connection between the studied problem and counterfactual regression or causal-based recommendation?

3. How to evaluate the performance of the proposed method in the case study presented in Section 5.2?

**Limitations:**

yes

**Strengths And Weaknesses:**

Strengths

1. It is sound to model causal matrix completion under multiple treatments as a tensor completion problem.

2. The paper analyzes the statistical properties of the method, including the finite-sample error bound, asymptotic normality, and the sample efficiency.

Weaknesses

1. In general, the contribution of the submission is incremental. The paper extends the existing work of causal matrix completion under binary treatment to the setting of multiple treatments. Mathematically, causal matrix completion is extended to causal tensor completion.

2. Assumption 2.5 is an important condition in the submission. It would be better to discuss more about the intuition behind the assumption. Especially, what is the physical meaning of the assumption in some specific application?

3. The related work discussion is inadequate. Many important references regarding causal inference and matrix/tensor completion are missing. In addition, the studied problem is highly relevant to counterfactual regression and causal-based recommendation. Such connections have not been well discussed.

4. The experiments could be improved. Although a case study is provided in Section 5.2, it is difficult to evaluate the performance of the proposed method since the ground-truth counterfactual results are unobserved. In addition, Section 5.2 does not present results of baselines for comparison, making it difficult to demonstrate the advantage of the proposed method.

5. For the experiments in Section 5.1, only one classic method is conducted for comparison. It would be better to conduct more baselines. I wonder if causal effect methods or matrix/tensor completion methods can be adopted for comparison.

---

> ### Author Rebuttal · Authors · 2026-03-31
>
> Thanks! Below we address the concerns point by point.
>
> **W1 (incremental contribution)** Thanks. We argue that a multi-treatment extension should be evaluated by whether it (i) recovers the previous binary setting as a special case, (ii) solves a new problem that the binary framework cannot handle, (iii) preserves the original statistical guarantees, and (iv) introduces genuinely new technical ideas rather than a formal 2D-to-3D reformulation. Our work satisfies all four. MSNN recovers SNN in the binary case, but more importantly, addresses the main challenge that, under multiple treatments, the observed data become ``intrinsically sparse`` rather than merely higher-dimensional. As a result, same-treatment anchors often do not exist, so a direct extension of binary causal matrix completion is generally not feasible.
>
> This sparsity is precisely what makes the problem nontrivial. Under the standard intuition, moving from binary treatment to multiple treatments should only worsen estimation, since the available data for each treatment level are fragmented and anchor matching becomes increasingly unlikely. Our result is counter to this default expectation. By establishing a new cross-treatment identification argument based on shared latent row factors, we show that the extra treatment dimension is not merely a source of sparsity, but can be turned into a source of information. This is the key technical insight behind MSNN. It enables mixed anchor construction across treatment levels and requires a new weighting scheme to handle treatment-specific scale and variance heterogeneity. As a result, MSNN not only makes estimation possible in regimes where SNN often fails, but also achieves ``exponential sample-efficiency improvement in sparse settings`` on both MCAR and MNAR data (see section 5.1.1 and reply to Q1 of reviewer jnH3 respectively), while still ``preserving the finite-sample and asymptotic guarantees of SNN``, mentioned by reviewers ``jnH3, ssQg, and RVA3``.
>
> **W2&Q1 (meaning of assumption 2.5)** Thank you for this important question. The intuition behind is that each unit (row) possesses ``intrinsic, stable characteristics`` that remain invariant regardless of the specific treatment (level $d$) applied.
>
> In our model $Y^{(d)} = U V^{(d)\top} + E^{(d)}$, the row factors $U$ represent the fundamental "profile" of the unit, while the treatment-specific loading matrices $V^{(d)}$ capture how the environment or intervention interacts with those profiles to produce an outcome.
>
> For example in E-commerce Pricing, consider "rows" as users and "treatments" as discount levels. The assumption suggests that a user’s underlying preference and price sensitivity traits are consistent. While a different level discounts (treatment) lead to different purchase probabilities, they are both acting upon the same latent user $u_i$.
>
> This assumption is the "bridge" that allows MSNN to aggregate information across treatments: if $U^{(d)}$ were treatment-dependent, observations from treatment $D_1$ would provide zero information about $D_2$. By sharing $U$, we effectively "borrow strength" from data-rich treatments to improve estimation in data-scarce ones.
>
> **W3&Q2 (missing related work)** Thank you for this point. We have enriched our Related Work to include the following perspectives and key references. Classical CFR methods [1] learn balanced representations over i.i.d. samples to estimate individual treatment effects. In contrast, our setting involves non-i.i.d., structured missingness over matrices/tensors. We extends the spirit of CFR to a low-rank completion regime, where balancing is achieved jointly with structural learning. Causal recommendation [2] treats exposure as treatment to debias MNAR user–item interactions. Our framework shares this view but addresses a more general matrix/tensor completion problem, leveraging structured missingness for joint debiasing and completion.
>
> [1] Johansson, Fredrik, Uri Shalit, and David Sontag. "Learning representations for counterfactual inference." International conference on machine learning. PMLR, 2016.
>
> [2] Bonner, Stephen, and Flavian Vasile. "Causal embeddings for recommendation." Proceedings of the 12th ACM conference on recommender systems. 2018.
>
> **W4&Q3 (evaluation of case study)** Thank you for the question. For the classic California Proposition99 study (section 5.2), although the ``counterfactual outcomes have no evaluation``, both the estimated pre-treatment outcomes and post-treatment outcomes of the same treatment with reality can be verified. Specifically, one can see ``how the dashed lines (estimation) fit the solid lines (observed data) of the same color``, verifying how our model fits real-world data.
>
> We have added the estimation of SNN as baseline, see https://anonymous.4open.science/r/ICML-rebuttal-MSNN-B460/dqF2/W4Q3/SNN_results/ . We observe that SNN infers less counterfactual potential outcomes than MSNN.
>
> (continue: response window of ssQg)

---

> > ### Author Rebuttal · Reviewer_dqF2 · 2026-04-04
> >
> > I appreciate the response. Most concerns have been addressed. However, such an extension from the binary setting to the multi-valued setting is not very intriguing to me. Many methods for binary treatments can be adjusted for multi-valued treatments without a significant modification. Here, the major advantage of the proposed method is based on the idea of shared latent row factors, which is quite common in collective matrix factorization and machine learning. I will not fight for rejection. But it is borderline to me.

---

> > > ### Author Response · Authors · 2026-04-04
> > >
> > > Thank you so much for your further comments. Again, thank you for your constructive questions, which help us improve our work. We respect your judgment and are grateful that you do not block acceptance. We have carefully and fully implemented your recommendation these days and included it in the current version.

---

### Official Review · Reviewer_jnH3 · 2026-03-13

**Soundness:** 3
**Presentation:** 2
**Significance:** 3
**Originality:** 3
**Overall Recommendation:** 4
**Confidence:** 3

**Summary:**

This paper proposes Mixed Synthetic Nearest Neighbors (MSNN) for causal matrix completion under multiple treatments with missing-not-at-random (MNAR) data. While the existing Synthetic Nearest Neighbors (SNN) method fails in data-scarce treatment levels due to strict same-treatment requirements, MSNN integrates information across treatments using mixed anchor sets, leveraging a shared latent row factor assumption. Theoretically, MSNN retains SNN's finite-sample error bounds and asymptotic normality while demonstrating an exponential improvement in sample efficiency for sparse treatments. Experiments on synthetic and real-world datasets validate its superior performance in data-limited regimes.

**Compliance With Llm Reviewing Policy:**

Affirmed.

**Key Questions For Authors:**

See Weaknesses.

**Limitations:**

Yes

**Strengths And Weaknesses:**

Strengths：
1. The paper successfully proves that MSNN retains the statistical properties of the original SNN algorithm, specifically the finite-sample error bounds and asymptotic normality.

2. The paper mathematically demonstrates the MSNN's sample efficiency improvement.

3. The synthetic experiments show that MSNN achieves strong empirical gains.

Weaknesses:

1. The theoretical proofs for sample efficiency are restricted to the Missing Completely at Random (MCAR) case. The paper currently lacks a theoretical guarantee that the exponential improvement in sample efficiency holds under MNAR.

2. In the proofs for the finite-sample bound (Theorem 4.5) and asymptotic normality (Theorem 4.6), $f(d)$ is treated as a known, deterministic constant. The authors need to either formally bound the estimation error of $\hat{f}(d)$ or clarify that the current theorems only hold in an oracle setting where treatment scales are perfectly known.

3. The entire cross-treatment identification strategy hinges entirely on Assumption 2.5, which dictates that the latent row factors $u_i$ are strictly invariant across all treatments. This is a remarkably strong structural assumption. Mathematically, it implies that no treatment can alter the dimensionality or the basis of the latent feature space. What happens if a severe or highly transformative treatment fundamentally changes a unit's behavioral pattern, thereby triggering a new latent dimension? I recommend that the authors include a robustness discussion (or empirical sensitivity analysis) detailing how the MSNN estimator degrades if Assumption 2.5 is moderately violated.

4. Missing Citations and Contextualization in the Introduction:

(i) In the opening paragraph, the authors state, "Recent advancements in matrix completion, a well-established tool from machine learning, have been adapted to this causal inference framework...". However, this claim is not supported by any citations. The authors should cite foundational works here to properly ground the statement.

(ii) The introduction mentions that "data are missing not at random (MNAR), as the very process of treatment assignment is often driven by unobserved latent factors". Given that MNAR is a central challenge addressed by this paper, it would be beneficial to cite 1-2 classic or highly relevant papers that tackle MNAR in causal inference or matrix completion to establish the significance of this problem.

(iii) When first mentioning the "Synthetic Nearest Neighbors (SNN) algorithm" by name, the authors should include an explicit citation to immediately orient the reader, rather than relying solely on the earlier mention in the text.

5. Several core assumptions and theorems are currently buried within the inline text, which disrupts the reading flow and diminishes their importance. The authors should use centered display equations for critical formulas.

---

> ### Author Rebuttal · Authors · 2026-03-31
>
> # A Kind General Response to All Reviewers and Dear AC
> We sincerely thank all reviewers for their thoughtful and encouraging feedback. We are glad that the paper was viewed as having **clear motivation** (``ssQg``), a **rigorous presentation and discussion of assumptions** (``ssQg``), and evaluation on a **widely used benchmark** (``ssQg``). We also appreciate that the methodology and claims were considered **sound** (``dqF2``) and **reasonably supported** (``RVA3``), and that the paper was recognized as providing a successful technical treatment together with **strong empirical gains** (``jnH3``). We are grateful for these positive assessments, and we address each comment carefully below.
>
> In this revision, we have carefully addressed all  concerns point by point, and summarized as below for quick comprehension:
>
> **(1) Strengthened theoretical results.**
>
> For guarantees beyond MCAR, we have extended our main theoretical result from MCAR to a general MNAR setting (``jnH3-W1``), showing that the exponential sample efficiency improvement of MSNN over SNN continues to hold. It clarifies how the result naturally recovers the MCAR case as a special instance.
>
> **(2) Expanded discussion and robustness of Assumption 2.5.**
>
> We have provided a clearer interpretation (stable unit-level latent traits),  justification via real-world examples (e.g., recommendation and policy analysis, ``dqF2-W2&Q1``, ``RVA3-Q1``), and both theoretical and empirical robustness analysis under controlled violations (``jnH3-W3``). Our results show that estimation error grows approximately linearly with perturbation magnitude, indicating robustness to moderate deviations from the assumption.
>
> **(3) Improved positioning with related literature and enhanced empirical evaluation/comparision.**
>
>  We have added baselines on classic nearest neighbor based MC methods on both synthetic experiments (``RVA3-Limitation 1``) and real-world case study (``dqF2-W4&Q3, RVA3``) to have better comparison. We have expanded the discussion connecting our work to Counterfactual Regression (CFR), Causal Recommendation, emphasizing the shared perspective of treating exposure as a causal mechanism (``dqF2-W3&Q2``).
>
> **(4) Improved presentation and clarity.**
>
>  We have reorganized theoretical sections, improved notation and readability, moved key inline equations to display form, and refined explanations throughout the paper.
>
> Overall, we believe the revised manuscript provides a clearer, more complete, and better-justified contribution, both theoretically and empirically. We are grateful for the reviewers’ insightful comments, which have significantly improved the work. We would be happy to further clarify any remaining questions.
>
> ---
> **Response to jnH3**
>
> Thanks! Below we address questions respectively.
>
> **W1 (lack of theoretical guarantee of sample efficiency under MNAR)**: Thanks! Under a generalized MNAR setting $p_{\rm MNAR}\left(A_{ij}^{d^\prime \in \mathcal{L}} , d \right) = \frac{e^{\lambda_d A_{ij}^{d} + \delta_d}}{p_0 + \sum_{d^\prime \ne 0} e^{\lambda_{d^\prime} A_{ij}^{d^\prime}  + \delta_{d^\prime}}}$, we have a similar result of exponential sample efficiency improvement:
>
> ``Theorem`` (informal). Consider estimation of entry $(i,j)$ under treatment $d$, uppose the treatment assignment follows the MNAR distribution above, for $\mathrm{(M)AR}(d) = r$ and $\mathrm{(M)AC}(d) = c$, under appropriate assumptions the sample efficiency  improvement is lower bounded by
> \begin{equation}
> \mathbb{E}[K_{\rm MSNN}(d)] / \mathbb{E}[K_{\rm SNN}(d)] \ge (1 - o(1))  \cdot ( 1 + (\sum_{d^{\prime} \in \mathcal{L} \backslash d} e^{(r+1)(-|\lambda_{d^{\prime}}| f(d^{\prime}) + \delta_{d^{\prime}})}) e^{(r+1)(-|\lambda_d| f(d) - \delta_d)} )^c .
> \end{equation}
>
> This reduces to the result under MCAR, by setting $\lambda_d = 0$ and $\delta_d = \ln p_d$ for all $d \in \mathcal{L}$. We have added this theorem to section 5.1.2 after MCAR (5.1.1) and the proof to Appendix.
>
> **W2 (lack of discussion in the estimation of $f(d)$)**: Thanks! Even though the foundational results require the perfect information on treatment scales in Theorem 4.5&4.6, we have to note that our result is **robust** when substituting $f(d)$ to its estimation $\hat{f(d)}$: even if the $\hat{f}(d)$s are estimated very roughly, the error bound for $A_{ij}(d)$ only increases from $f(d) \cdot ...$ to $\hat{f}(d) \max_{d^\prime \in \mathcal{L}} f(d^\prime)/\hat{f}(d^\prime)$. We have added it at the end of section 4.2.
>
> **W3 (lack of discussion in violation of Assumption 2.5)**: Thank you for pointing out this critical point. We provide robustness analysis both theoretically and empirically.
> Suppose latent row factors $U^{(d)}$ are perturbated by noise level $\zeta \in \mathbb{R}$, i.e. $U^{(d)} = U + \zeta U_{\rm perturb}^{(d)}$, where $U$ is shared and $U_{\rm perturb}^{(d)}$ is noise. Larger $\zeta$ indicates stronger perturbation.
>
> (continue: response window of ssQg)

---

> > ### Author Rebuttal · Reviewer_jnH3 · 2026-04-05
> >
> > I thank the authors for their rebuttal and will keep my score.

---

### Decision · Program_Chairs · 2026-04-30

**Decision:**

Accept (regular)

**Comment:**

This paper provides finite sample error rates and asymptotic normality results for a generalization of the causal matrix completion model from [1] where there are multiple treatment types. The low-rank assumption is replaced by a Tucker 2 condition and the original synthetic nearest neighbor algorithm is updated to a proposed “Mixed synthetic nearest neighbor algorithm” which similarly exploits the assumption of sufficiently many anchor rows and columns.



The reviewers are generally **mildly positive** about the paper, praising the novelty of the results and general. However, there are multiple concerns about the use of multiple unintuitive assumptions and the lack of technical innovation in the proofs. In particular, reviewer RVA3 mentions that some of the results are known and the authors do not clearly delineate which parts of the proof are original. In a private reviewer conversation, the *positive reviewer ssQg also admits that the extension is incremental*, which agrees with the statement of reviewer dqF2. Reviewer jnH3 also mentions that the submission only covers the MCAR case, to which the authors provide an informal theorem statement extending to the general case. Reviewer RVA3 also complains of insufficient polishing.

This is certainly a **borderline paper**. Although the scores are unenthusiastically positive, the overall sentiment in the text appears negative. The results are interesting but incremental and unpolished. In particular, the assumptions are not properly discussed. From a quick look at the paper, I agree that the polishing is insufficient and the submission seems rushed. The main assumption is not even connected to the concept of Tucker 2 decomposition, making it more difficult to read on the first reading. The fact that the tucker 3 cas isn't covered is also an issue. However, the paper makes a technically correct and technically original, if incremental, contribution. Therefore, I would still lean towards acceptance. Given most of the lemmas are known and the technical difficulty is limited, I believe it is reasonable to expect the authors to **cover the MNAR case** as promised in the rebuttal and to **improve polishing** and **expand the experimental section** (as requested by reviewer dqF2) in the camera-ready revision.



**References**

[1] Causal Matrix Completion. Anish Agarwal, Munther Dahleh, Devavrat Shah, Dennis Shen.